



# HERMESv3, a stand-alone multiscale atmospheric emission modelling framework - Part 2: bottom-up module.

Marc Guevara[1], Carles Tena[1], Manuel Porquet[1, *], Oriol Jorba[1], Carlos Pérez García-Pando[1]

[1]Earth Sciences Department, Barcelona Supercomputing Center, Barcelona, 08034, Spain
[*]now at: Departamento de Geografía y Ordenación del Territorio, Universidad de Zaragoza, Zaragoza, 50009, Spain

*Correspondence to*: Marc Guevara (marc.guevara@bsc.es)

**Abstract.** We describe the bottom-up module of the High-Elective Resolution Modelling Emission System version 3 (HERMESv3), a python-based and multiscale modelling tool intended for the processing and computation of atmospheric emissions for air quality modelling. HERMESv3 is composed of two separate modules: the *global_regional* module and the
*bottom_up* module. In a companion paper (Part 1, Guevara et al., 2019) we presented the *global_regional* module. The *bottom_up* module described in this contribution is an emission model that estimates anthropogenic emissions at high spatial (e.g. road link level,) and temporal (hourly) resolution using state-of-the-art calculation methods that combine local activity and emission factors along with meteorological data. The model computes bottom-up emissions from point sources, road transport, residential and commercial combustion, other mobile sources and agricultural activities. The computed pollutants
include main criteria pollutants (i.e. $NO_x$, CO, NMVOC, $SO_x$, $NH_3$, $PM_{10}$ and $PM_{2.5}$) and greenhouse gases (i.e. $CO_2$ and $CH_4$, only related to combustion processes). Specific emission estimation methodologies are provided for each source, and are mostly based on (but not limited to) the calculation methodologies reported by the European EMEP/EEA air pollutant emission inventory guidebook. Meteorological-dependent functions are also included to take into account the dynamical component of the emission processes. The model also provides several functionalities for automatically manipulating and performing spatial
operations on georeferenced objects (shapefiles and raster files). The model is designed so that it can be applicable to any European country/region where the required input data is available. As in the case of the *global_regional* module, emissions can be estimated on several user-defined grids, mapped to multiple chemical mechanisms and adapted to the input requirements of different atmospheric chemistry models (CMAQ, WRF-Chem and MONARCH) as well as a street-level dispersion model (R-LINE). Specific emission outputs generated by the model are presented and discussed to illustrate its capabilities.



# 1 Introduction

The development of reliable anthropogenic emission inventories is crucial to understand air pollution sources and design effective emission abatement measures (Day et al., 2019). Emission inventories are also key inputs for air quality research and forecasting applications, and represent one of the largest source of uncertainty in the air quality modelling chain (Russell and Dennis, 2000). For their use in atmospheric chemistry models, emissions need to be spatially distributed over a gridded domain, temporally resolved with (typically) hourly intervals and mapped to the species defined in the gas phase and aerosol chemical mechanisms of the atmospheric chemistry model.

The last decades have seen large efforts to develop emission inventories for global and regional scales using a variety of input datasets and approaches. These inventories have become key in supporting scientific research and policy-making (e.g. the Air Quality Modelling Evaluation International Initiative, AQMEII; Pouliot et al., 2015). At global scale, some of the most frequently used inventories are the Air Pollutants and Greenhouse Gases Emission Database for Global Atmospheric Research (EDGAR, Cripa et al., 2018) and the dataset derived from the Evaluating the Climate and Air Quality Impacts of Short-Lived Pollutants project (ECLIPSEv5.a, Klimont et al., 2017). There are also widely known regional emission inventories such as the European Monitoring and Evaluation Programme (EMEP) (Mareckova et al., 2019) and the TNO-MACC_III (Kuenen et al., 2014) or the Regional Emissions inventory in Asia (REAS), which covers China, Japan and other Asian countries (Kurokawa et al., 2013). More recently, and as part of the European Copernicus Atmosphere Monitoring Service (CAMS), new global and regional emission datasets covering both anthropogenic and natural sources have been developed (Granier et al., 2019).

These inventories provide estimates of emissions either globally or regionally for a variety of sectors, pollutants and years in a consistent way. However, they are usually limited for high resolution modelling applications; assessing urban air quality or the local impact of emission reduction measures are two good examples (e.g. (Timmermans et al., 2013). Such a limitation is due to the insufficient level of detail of the data used to estimate the emissions, typically national statistics, the uncertainties associated to their spatial and temporal distribution and the lack of flexibility for computing specific scenarios (e.g change of speed limits). Emissions are first estimated at the annual and national level and the spatial proxies assigned to each pollutant source (e.g. population, city lights, land uses) are usually empirical and may not be representative of the real-world spatial emission patterns (Andres et al., 2016; Geng et al., 2017). Hourly emissions are computed through the application of temporal profiles (i.e. monthly, weekly and diurnal) to the original annual inventories, which are usually static (i.e. spatially constant) and do not account for the dynamical component of the emission processes (e.g. volatilization of ammonia as a function of meteorological parameters; Backes et al., 2016). Moreover, existing global and regional emission inventories are reported following sector aggregations (e.g. Gridded Nomenclature For Reporting, GNFR) that can hinder the application of detailed speciation profiles. For instance, the EMEP inventory, which compile emissions from the parties of the Convention on Long-



range and Trans-boundary Air Pollution (CLRTAP), reports all road transport emissions under one single category (i.e. GNFR F_RoadTransport) without discriminating by vehicle type (e.g. passenger cars, heavy duty vehicles), fuel type (i.e. petrol and diesel vehicles) or EURO category. These factors become crucial when assigning the fraction of total nitrogen oxides ($NO_x$) emitted directly as nitrogen dioxide ($NO_2$) (e.g. Carslaw and Rhys-Tyler, 2013).

Dedicated emission models combining detailed databases with novel calculation methods that represent the main factors influencing the emission processes (e.g. meteorology, soil properties) can overcome these limitations. Some recent examples are the French VOLT'AIR model (Hamaoui-Laguel et al., 2014), which simulates ammonia ($NH_3$) volatilisation fluxes after the application of fertiliser taking into account agro-environmental factors (e.g. meteorology, agricultural practices, soil

properties), the Brazilian VEIN model (Ibarra-Espinosa et al., 2018), which provides bottom-up exhaust and evaporative vehicular emissions at street and hourly levels using different sets of emission factors (e.g. COPERT, EPA), or the Norwegian MetVed model (Grythe et al., 2019), which estimates residential wood combustion hourly emissions on a 250m grid resolution considering several influencing factors such as outdoor temperature, type of available heating technologies and the number of and type and size of dwellings.

Despite presenting highly accurate and detailed modelling methodologies, there are still some shortcomings associated to these tools, mainly in terms of their usability for air quality modelling. On the one hand, each model covers only a specific pollutant sector, which means that they may have to be manually combined with other existing inventories. On the other hand, the models are usually designed to provide emissions for specific frameworks, which results in limitations in terms of number of

atmospheric chemistry models compatible with the emission outputs, map projections and type of working domains supported, as well as regions/countries in which the model can be applied. Finally, these tools are not always distributed under open access licenses, which limits their usage within the scientific community.

This paper is the second part of the description of an open-source, python-based, parallel, stand-alone and multiscale

atmospheric emission modelling framework. The High-Elective Resolution Modelling Emission System version 3 (HERMESv3) estimates atmospheric emissions for use in multiple air quality models (i.e. CMAQ, Appel et al., 2017; WRF-Chem, Grell et al., 2005 and NMMB-MONARCH, Badia et al., 2017) as well as map projections and model grids (i.e. regular and rotated latitude-longitude, lambert conformal conic, mercator). The system reflects the learning from previous versions of HERMES developed by the Earth Sciences Department of the Barcelona Supercomputing Center (BSC) during the last decade

(i.e. Baldasano et al, 2008; Ferreira et al., 2013; Guevara et al., 2013 and 2017).

HERMESv3 is composed of two independent modules named *global_regional* and *bottom_up*. The *global_regional* module (HERMESv3_GR) is a customizable emission processing system that combines existing gridded inventories with user-defined



vertical, temporal and speciation profiles for the generation of global and regional air quality model-ready emission files. A complete description of HERMESv3_GR can be found in Guevara et al. (2019).

The *bottom_up* module (described in this paper and further referred to as HERMESv3_BU) is an emission model that computes high spatial (e.g. road link, point source) and temporal (i.e. hourly) resolution anthropogenic emissions using state-of-the-art calculation methods that combine local activity and emission factors along with meteorological data. The model covers the estimation of bottom-up emissions from multiple sources, including power and manufacturing industries, road transport (i.e. exhaust and non-exhaust sources), residential and commercial combustion, other mobile sources (i.e. agricultural machinery, landing and take-off cycles in airports, shipping activities in ports and recreational boats) and agricultural activities (manure management, fertilizer application and crop operations). The computed pollutants include main criteria pollutants (i.e. $NO_x$; CO; NMVOC; $SO_x$; $NH_3$; $PM_{10}$ and $PM_{2.5}$) and greenhouse gases (i.e. $CO_2$ and $CH_4$, only related to combustion processes). HERMESv3_BU provides specific estimation methodologies and emission factors for each source, which are mostly based on (but not limited to) the calculation methodologies reported by the European EMEP/EEA air pollutant emission inventory guidebook (EMEP/EEA, 2016). Users are allowed to load their own emission factors or apply specific tuning factors to the default dataset. With respect to the input activity data (e.g. geographic location of the industrial facilities and corresponding activity factors and temporal profiles) the user is responsible for providing the required information of the region of interest following the formats established in HERMESv3_BU. The application of the model is not restricted to a specific country; it can be run for any European region if the corresponding input data is provided. HERMESv3_BU includes a variety of global and regional state-of-the-art datasets to increase the usability of the tool and minimize the amount of input information that needs to be provided by the user. Following the same line of thinking, HERMESv3_BU includes functionalities similar to Geographic Information Systems (GIS) for automatically manipulating and performing spatial operations on geometric objects (e.g. remap spatial data from one spatial domain to another). The model counts as well with a flexible speciation mapping functionality, which allows the user to speciate the original pollutants to any desired chemical mechanism. Besides the aforementioned mesoscale atmospheric chemistry models that are compatible with HERMESv3, the road transport emission outputs of HERMESv3_BU can also be used within the R-LINE research grade dispersion model (Snyder et al., 2013).

## 2    General description

### 2.1    Overview

A schematic of the model structure and execution workflow is shown in Fig. 1. The characteristics of the working domain, execution dates, emission sectors and pollutants to include in the calculation process, input data paths, atmospheric chemistry output format and number of calculation and writing processors have to be specified by the user in the general configuration file (see section 2.2). Once all this information is compiled, HERMESv3_BU starts the general initialization process, which

includes: (i) the creation of the working grid where emissions will be calculated (grid function), (ii) the creation of the polygon feature that will be used to perform clip operations of the input data when needed (clip function) and (iii) the distribution of the computational resources among the different emission sectors (sector manager function). The data generated with the grid and clip functions, which are specific of each working domain, are stored as auxiliary files by default after their creation so that they can be reused in subsequent executions.

The input data used by HERMESv3_BU includes sector-specific and general data. The first set of information is divided in individual folders (<sector>), each one containing the activity and emission factor files of each sector, and the <profiles> folder, which contains the temporal, vertical and speciation profiles associated to each sector. Regarding the general data, this section includes meteorological files, which are used by several sectors, as well as global and regional datasets such as population maps. The different type of files used in the model are described in detail in section 3.

The emission core of the model is composed of individual and independent submodules that calculate the emissions of each sector following specific estimation methodologies (see sections 3.1 to 3.6). Most of the submodules share a common procedure, which consist on the following steps: (i) sector initialization, which creates sector-specific auxiliary files, (ii) sector calculation, (iii) spatial mapping, (iv) temporal distribution and (v) speciation.

Once the execution of all the sectors is finished, HERMESv3_BU starts the data writing process. This function consists on first gathering the emissions estimated by each submodule (provided as 4D matrices with information of emissions across space, time and vertical levels) and secondly writing the merged data in an output NetCDF file according to the conventions of the atmospheric chemistry model of interest (see section 2.5).

## 2.2 General configuration file

The general configuration options are passed to HERMESv3_BU via a configuration file, which is divided into seven different sections.

1. General: this section defines the main paths of the model (i.e. input, output, general data, auxiliary files), the name of the output emission file, time step configuration parameters (i.e. start and end dates and number of hourly time steps) and the option to remove the existing auxiliary files at the beginning of the execution.

2. Domain and output format: this section defines the characteristics of the working grid (e.g. spatial coverage, horizontal resolution and vertical description) as well as the conventions of the output NetCDF emission file. Currently, HERMESv3_BU supports four map projections (i.e. regular lat–long, rotated lat–long, Lambert conformal conic and Mercator) and three atmospheric chemistry model file conventions (i.e. CMAQ, WRF-Chem and NMMB-MONARCH) (see section 2.5).





3. Clipping: This section defines the polygon feature that will be used to perform the clipping operation during the general initialization process (see section 2.4).

4. Sector management: This section defines the number of computational processors that will be assigned to each pollutant sector during the emission calculation process. Sectors can be individually deactivated setting their corresponding numbers to 0. This section also defines the number of processors that will be assigned for the writing process (see section 4).

5. General shapefiles and raster: This section define the path to the general shapefiles (i.e. administrative boundaries) and rasters (i.e. population, land use, livestock and soil property maps) used in the model.

6. Pollutant sector data: This section contains individual subsections for each pollutant sector, in which the user defines: (i) the list of pollutants to be calculated, (ii) the data paths that point to the specific-sector information used for the emission calculation process and (iii) (only available for certain sectors) an optional subset of pollutant categories to be considered for the calculation process (i.e. list of vehicle categories in the road transport sector, list of crop categories in the fertilizer application sector, list of animal categories in the livestock sector, list of airport codes in the aircraft sector, list of port codes in the shipping sector, list of fuels in the residential and commercial sector). This last option can be very useful when studying the contribution of certain pollutant categories to total sectoral emissions (e.g. diesel vehicles in road transport) or when performing source attribution modelling studies (e.g. Pay et al., 2019).

7. Meteorology: This section define the paths to the gridded meteorological files used as input.

## 2.3 Input files and cross-referencing

HERMESv3_BU uses four types of input file formats:

- ESRI shapefiles (points, polygons and polylines): Used to provide spatial georeferenced information, including road transport networks, collections of point source facilities, infrastructure boundaries (i.e. airport, port) and administrative boundaries.

- Geotiff raster files: Used to provide spatially gridded information, including land use information, population and livestock distributions and soil properties (i.e. pH and cation exchange capacity).

- CSV files: Used to store non-georeferenced activity data and emission factors as well as sets of temporal, vertical and speciation profiles.

- NetCDF files: Used to provide modelled meteorological data (e.g. temperature, wind speed). HERMESv3_BU can currently use gridded meteorological data provided by NMMB-MONARCH and ERA5 (C3S, 2017).

Depending on the pollutant sector one or more types of data are combined during the emission calculation process. For each sector, cross-references and spatial relationships between files are used for matching all the different input data. Figure 2 shows an example of the files used in the point source sector (see section 3.1) and how all the information is linked. Most of the point source's input information (e.g. activity and emission factors, stack parameters, geographical coordinates) used by the model





is provided in a multipoint shapefile, each row containing the information of a specific facility. The shapefile includes temporal and speciation profile IDs (e.g. "MXXX" for the monthly profiles, "XXX" being a three-digit numeric code that starts at "001") which are cross-referenced with temporal and speciation CSV files where the numeric profiles are stored. On the other hand, the geographical coordinates of each facility are used to identify the closest grid cell of the meteorological NetCDF file

and subsequently associate to them the required meteorological information.

The input data required to run HERMESv3_BU can be classified into three main categories:

- User-dependent data files: Files that contain local information for the domain of study (e.g. energy consumption statistics, cultivated crop areas) and that need to be provided by the user.

- Built-in data files: Store information that is not tied to a specific domain (e.g. emission factors, temporal profiles) and that is provided by default with the model.  Users can modify the files provided by default if needed or add new ones (e.g. speciation profiles for a new chemical mechanism) but it is not mandatory in order to correctly run the model.

- External data files: Open source files reported by third parties (e.g. Joint Research Centre, Copernicus Land Monitoring Service) that contain global or regional information (e.g. population density, land use map) and that allow

minimizing the amount of local information that needs to be provided by the user

A classification of all the needed input files according to the three categories described above can be found in Table B1. Section 3 provides more information on the input datasets used for each pollutant sector. A complete description of all the input files and corresponding information fields can be found in the wiki of the model (see section 6).

**2.4     Spatial operations**

HERMESv3_BU includes multiple functionalities for manipulating and performing spatial operations on geometric objects that are spatially referenced and have associated attributes (i.e. shapefiles and raster files) without requiring the users to have a Geographic Information System (GIS) software. This allows the model to automatically manipulate georeferenced input data files, as well as to create spatial surrogates that are lately used to map the estimated emissions onto the grid cells of the desired

working domain. The following operations are implemented in the model:

- Read/Write/Create: Reads, writes and creates vector-based spatial data including ESRI shapefiles and Geotiff raster files. These operations also allow changing the map projection of the original files.

- Clip: Overlays a polygon on a target feature and extracts from it only the data that lies within the area outlined by the

clip polygon. By default, the clip polygon is defined as the outline of the user-defined working domain, but users can optionally use an existing shapefile (e.g. administrative boundary) or define a costume polygon providing a set of latitude-longitude coordinates. The clipped data becomes a new feature.



- Unary union: Returns a representation of the union of the given geometric objects. This function is used to create the outline of the user-defined gridded working domain.

- Data conversion: Converts raster files to a polygon feature class.

- Spatial intersection: Computes a geometric intersection of two features. The operation returns only those geometries that are contained by both targeted features. Unlike the clip operation, in the spatial intersection the associated attribute values from the input feature classes are copied to the output feature class.

- Spatial difference: As opposed to the spatial intersection, in this case the operation returns only those geometries that are not contained by both targeted features.

- Spatial join: Joins attributes from one feature to another based on the spatial relationship. The target features and the joined attributes from the join features are written to the output feature class. Unlike the spatial intersection, the spatial join does not modify the geometry of the target feature.

- Nearest point: Calculate the nearest point in a pair of geometries. This operation is used to assign to each emission source (e.g. point source, road link) the closest meteorological data reported in the NetCDF input files.

As an illustration, Figure 3 shows the steps performed by HERMESv3_BU to generate the gridded fuel consumption data used by the residential and commercial combustion emission submodule (see section 3.4). In the example, Spanish natural gas and wood consumption data obtained at the province level (IDAE, 2018; MITECO, 2018) are mapped onto a 4km by 4km regional lambert conformal conic grid covering the Iberian Peninsula. For creating the gridded data, the model uses the population maps reported by the Global Human Settlement Layer (GHSL) project (Florczyk et al., 2019). The GHSL provides global Geotiff

raster files at a resolution of 1km by 1km on the distribution and density of population, expressed as the number of people per cell (Schiavina et al., 2019) and on the classification of human settlements on the base of the built-up and population density, expressed as high and low density clusters (i.e. large and small urban areas, here remapped under a single category expressed as urban areas) and rural areas (Pesaresi et al., 2019). In the example, a clip of the original GHSL population density raster is performed using a shapefile of the administrative borders of Spain (Figure 3.a). The resulting clipped raster is converted to a

polygon feature (Figure 3.b, zoom over the region of Madrid) to which new information is appended performing two spatial joins: one with a shapefile of the Spanish Nomenclature of Territorial Units for Statistics level 3 (NUTS3) administrative boundaries to append the province code to each source grid cell, and another one with the GHSL settlement classification layer (that has also been previously converted from raster to shapefile) to append the population type information (Figure 3.c). Once each grid cell of the polygon has information on the population, the NUTS3 code and type of settlement, HERMESv3_BU

spatially distributes the annual fuel consumption input data, which is provided by the user in a CSV file. For that, the following expression is applied (Eq. 1):





$$FC_f(\bar{x}) = FC_{f,n} * \frac{Pop_{n,t}(\bar{x})}{\sum_{x=1}^{N} Pop_{n,t}(\bar{x})} \tag{1}$$

Where $FC_f(\bar{x})$ is the annual fuel consumption [GJ year$^{-1}$] of fuel $f$ on the source grid cell $\bar{x}$; $FC_{f,n}$ is the total annual fuel consumption [GJ year$^{-1}$] of fuel $f$ in the NUTS3 $n$ and $Pop(\bar{x})_n$ is the amount of population [inhabitant cell$^{-1}$] of type $t$ (urban, rural) from NUTS3 $n$ on the source grid cell $\bar{x}$. In the example provided, urban and rural population are considered for the

distribution of natural gas consumption (Figure 3.d), and only rural population for the distribution of wood consumption (Figure 3.e).

In the final step, the resulting polygon features are spatially intersected with the 4km by 4km gridded domain in order to remap the fuel consumption data from the source domain to the destination domain (Figure 3.f and Figure 3.g). The remapping is

performed taking into account the ratio of the area of the region of intersection between the source and destination grid cells $(A(\bar{x}, x))$ to the total area of the source grid cell $(A(\bar{x}))$, as expressed in Eq. (2):

$$FC_f(x) = FC_f(\bar{x}) * \frac{A(\bar{x}, x)}{A(\bar{x})} \tag{2}$$

Similar operations are applied for the spatial manipulation of other georeferenced datasets such as land use categories, livestock

maps or digitalised traffic networks.

## 2.5 Air quality model-ready files

HERMESv3_BU is prepared to create NetCDF emission files following the conventions of multiple air quality models, including CMAQ, WRF-Chem and NMMB-MONARCH. For each model, an independent writing function has been implemented (e.g. *writing_cmaq.py*) to perform the required conversion of units and inclusion of mandatory global attributes.

This modular approach allows to easily extend the writing capabilities of the model to other atmospheric chemistry model conventions. Alternatively, the user can also estimate the emissions in a so-called DEFAULT format, which stores the emissions [g·h$^{-1}$] in a NetCDF file that follows the Climate and Forecast (CF1.6) Metadata Conventions. In the case of road transport emissions, an alternative writing function was designed so that the computed link-level emissions can be used by the R-LINE Gaussian dispersion model. This functionality allows HERMESv3_BU to be used for modelling air pollution at the

urban (street level) scale (Benavides et al., 2019).



## 3 Emission sectors

The following subsections provide a detailed description of the emission estimation methodologies implemented within HERMESv3_BU for each pollutant sector. Unless otherwise stated, all the equations reported in the following subsections are derived from the emission estimation expressions reported by EMEP/EEA (2016). Original expressions have been

reformulated to produce high resolution emissions (i.e. gridded or source-specific and hourly) instead of total national annual emissions. Some illustrative examples of the outputs that can be generated with the tool are also presented and compared against other existing emission datasets. In all the cases, the presented results were estimated for Spain or a Spanish region/city. The reason for this is mainly related to the access to the required local, regional and national data that the authors have for this country. Compiling data with the same level of detail for other countries is out of the scope of this work. Nevertheless, and as

mentioned before, HERMESv3_BU is designed so that it can be applicable to other European countries/regions where similar input data is available.

### 3.1 Point sources

This submodule estimates hourly emissions from process and combustion activities occurring in energy and manufacturing industrial point sources Eq. (3):

$$E_{p,i}(h) = AF_p * EF_{p,i} * FM(m)_p * FW(d)_p * FH(h)_p \qquad (3)$$

Where $E_{p,i}(h)$ are the hourly emissions of pollutant $i$ at point source $p$ and hour $h$ [g h$^{-1}$]; $AF_p$ is the annual activity factor (energy/material produced, fuel consumed) associated to point source $p$ [GWh year$^{-1}$ or GJ year$^{-1}$ or g of product year$^{-1}$]; $EF_{p,i}$ is the emission factor linked to point source $p$ and pollutant $i$ [g GWh$^{-1}$ or g GJ$^{-1}$ or g g of product$^{-1}$]; $FM(m)_p$ is the monthly

factor associated to month $m$ and point source $p$ [0:1]; $FW(d)_p$ is the weekly factor associated to day $d$ and point source $p$ [0:1] and $FH(h)_p$ is the hourly factor associated to hour $h$ and point source $p$ [0:1].

As previously mentioned, most of the input data is provided in a georeferenced multipoint shapefile, each row containing the information for each specific facility (see section 2.4). Emission factors are derived from facility-level emission reports when

available, as recommended by the Tier 3 approach of EMEP/EEA (2016) (Chapters 1.A.1 and 1.A.2). Alternatively, Tier 2 technology and fuel dependent emission factors provided by the European guidelines are proposed. For each point source, emissions are horizontally allocated to the nearest grid cell of the destination working domain. Regarding the vertical allocation, HERMESv3_BU explicitly uses plume rise calculations to determine for each hour and each point source the effective emission heights. For this, the plume rise formulas as described by Gordon et al. (2018) are implemented. The

algorithm takes into account stack and meteorological parameters, including: stack height, stack diameter, exit temperature at the stack outlet, stack emission exhaust velocity, air temperature at stack height, wind speed at stack height, surface



temperature, boundary-layer height, friction velocity and Obukhov length. Emissions are uniformly allocated across all the vertical layers that are included between the top and the bottom of the calculated plume. The plume rise function can be deactivated in HERMESv3_BU. In that case, the model will allocate the emissions to the layer closest to the stack height.

Alternatively to Eq. (3), HERMESv3_BU can directly ingest measured hourly emissions if available. For this, the user needs to provide a separate CSV file that contains the point source's measured emission fluxes per hour of the day [g h⁻¹] and to define the $AF_p$ parameter in the shapefile as '-1'. This functionality becomes very relevant when assessing the impact of point source's plumes for specific days and under specific meteorological conditions (e.g. Baldasano et al., 2014).

Figure 4 shows an example of hourly and vertically distributed $SO_2$ emissions [kg h⁻¹] estimated by HERMESv3_BU for the As Pontes coal-fired power plant (Spain) during the months of January and July 2015. As Pontes is the largest power plant in Spain (1468.5 MW), and its exhaust stack (356 m) is the largest in the country and the second largest in Europe. The emission fluxes are directly derived from measurements reported by the Spanish Research Centre for Energy, Environment and Technology (CIEMAT, personal communication). The meteorological parameters for the plume rise calculations are obtained

from the NMMB-MONARCH model. It can be seen that there are significant differences between the vertical profiles obtained for January (winter) and July (summer), the emissions being injected to lower altitudes in the first case. The average plume thickness and plume top in January is 219.3m and 685.5m, respectively, while in July the values are 259.4m and 745.7m (18.3% and 8.8% larger, respectively). This is mainly due to meteorological differences between July and January in terms of air temperature at the stack height (+6.5 °C), and boundary-layer height (-430.5m). The results are in line with other plume

rise calculations performed in other facilities located in similar climate zones (Bieser et al., 2011).

### 3.2    Road transport

#### 3.2.1    Hot and cold exhaust

Hot exhaust emissions are estimated following Eq. (4):

$$Ehot_{l,i}(h) = \sum_{v=1}^{n} AADT_{v,l} * L_l * EFhot(V(h)_l)_{v,i} * Mcorr(V(h)_l)_v * FM(m)_l * FW(d)_l * FH(h)_l \tag{4}$$

Where $Ehot_{l,i}(h)$ are the hourly hot exhaust emissions of pollutant $i$ at road link $l$ and hour $h$ [g h⁻¹]; $AADT_{v,l}$ is the annual average daily traffic for vehicle category $v$ at road link $l$ [n° vehicles day⁻¹]; $L_l$ is the length of the road link $l$ [km]; $EF(V(h)_l)_{v,i}$ is the hot exhaust emission factor linked to vehicle category $v$ and pollutant $i$ [g km⁻¹ n° vehicles⁻¹] as a function of the hourly mean vehicle travelling speed $(V(h)_l$ at road link $l$ and hour $h$ [km h⁻¹]; $Mcorr(V(h)_l)_v$ is the mileage correction factor

associated to vehicle category $v$ also estimated as a function of the hourly mean travelling speed; $FM(m)_l$ is the monthly factor





associated to month $m$ and road link $l$ [0:12]; $FW(d)_l$ is the weekly factor associated to day $d$ and link $l$ [0:1] and $FH(h)_l$ is the hourly factor associated to hour $h$ and link $l$ [0:1]. The number of vehicle categories is $n$.

Most of the activity input data (e.g. average daily traffic flow, mean vehicle speed) is provided in a multiline shapefile, each

5 row containing the information of a specific road link. The shapefile includes vehicle fleet composition, temporal and speciation profile IDs, which are cross-referenced with the corresponding CSV files where all the numeric profiles are stored (similarly to the example shown for point sources, see section 2.3). These profiles are used to distribute the total traffic flow among the different vehicle categories, temporally disaggregate the traffic flow and average speed at the hourly level and speciate the estimated emissions, respectively.

Both the emission and mileage correction factors implemented in HERMESv3_BU are the ones reported by the Tier 3 methodology of EMEP/EEA (2016) (Chapter 1.A.3.b.i-iv), which correspond to the values reported by the European COmputer Programme to calculate Emissions from Road Transport version 5 (COPERT 5; https://copert.emisia.com/). A total of 491 vehicle categories are considered, discriminated by vehicle type (i.e. mopeds, motorcycles, passenger cars, light duty

15 vehicles, heavy duty vehicles, buses), fuel type (i.e. diesel, gasoline, LPG, hybrid, electric), EURO category, engine power and gross weight class.

In the case of cold-start emissions, the calculation expression used is the following one (Eq. 5):

$$
Ecold_{l,i}(h) = \sum_{v=1}^{n} Ehot_{l,i,v}(h) * \beta_{i,v}(T(h)_l) * \left( Qcold_{l,i,v}(V(h)_l, T(h)_l) - 1 \right) \tag{5}
$$

Where $Ecold_{l,i}(h)$ are the hourly cold exhaust emissions of pollutant $i$ at road link $l$ and hour $h$ [g h$^{-1}$]; $Ehot_{l,i,v}(h)$ are the hourly hot exhaust emissions of pollutant $i$ for vehicle category $vi$ at road link $l$ and hour $h$ [g h$^{-1}$]; $\beta_{i,v}(T(h)_l)$ is the fraction of mileage driven with a cold engine pollutant $i$ and vehicle category $k$ [0:1] as a function of the hourly outdoor temperature $T(h)_l$ for hour $h$ and road link $l$ [ºC] and $Qcold_{l,i,v}(V(h)_l, T(h)_l)$ is the cold/hot emission quotient for pollutant $i$ and vehicles

25 category $v$ [$\geq 1$] as a function of the hourly outdoor temperature $T(h)_l$ [ºC] and hourly mean vehicle travelling speed $(V(h)_l$ for hour $h$ and road link $l$ [km h$^{-1}$]. The number of vehicle categories is $n$.

As in the case of the hot exhaust emissions, the $\beta_{i,v}(T(h)_l)$ and $Qcold_{l,i,v}(V(h)_l, T(h)_l)$ parameters are estimated following the expressions and constants reported by the EMEP/EEA (2016) Tier 3 methodology (Chapter 1.A.3.b.i-iv).





Besides the COPERT 5 constants used to calculate vehicle and pollutant specific hot and cold emission factors, HERMESv3_BU also includes scaling factor parameters (defined as 1 by default) that the user can modify to tune the original emission factors. This functionality can be useful for adjusting the default factors based on the insights reported by measurements performed under real world driving conditions (e.g. underestimation of COPERT $NH_3$ cold start emissions

according to Suarez-Bertoa et al., 2017). The mileage correction factors reported by COPERT 5, which only applies to gasoline vehicles, are expanded to diesel vehicles (i.e. deterioration of tailpipe $NO_x$ emissions of 22% and 10% on EURO 2 and 3 diesel passenger cars), following the results reported by Chen and Borken-Kleefeld (2016).

The estimated link-level vehicle emissions are mapped onto the user-defined gridded working domain by applying a spatial

intersection. Once the intersection is performed, emissions are automatically gathered at the grid cell level and the total sum is computed. Figure 5 shows an example of the hourly PM2.5 road transport emissions estimated for an area of Barcelona city (09:00h UTC), both at the road link level (kg km$^{-1}$ h$^{-1}$, Figure 5.a) and grid cell level (1kmx1km) (kg h$^{-1}$, Figure 5.b). Total annual $NO_x$ and PM10 road transport emissions were estimated for the city of Barcelona using HERMESv3_BU and the results were compared against the last available local emission inventory developed by the Barcelona City Council (AB, 2015) (Figure

5.c). Information on the traffic flow data was obtained from the local automatic traffic counting network (Barcelona city council, mobility and transport department, personal communication) and the TomTom historical average speed profiles product ([https://www.tomtom.com](https://www.tomtom.com)), whereas vehicle fleet composition profiles were derived from a remote sensing campaign (RACC, 2017). It is observed that HERMESv3_BU results are 60% and 39% higher than the ones reported by the AB. This is due to a combination of several factors, including: (i) the different years of reference (2017 for HERMESv3_BU and 2013 for

AB), (ii) the inclusion of the Barcelona's port area associated road transport in HERMESv3_BU (large amount of heavy duty vehicles that contribute with more than 250 and 15 t year$^{-1}$ of $NO_x$ and PM10, respectively), (iii) the use of COPERT 5 real-world adjusted $NO_x$ emission factors for EURO 5 and 6 diesel vehicles in HERMESv3_BU (AB inventory is based on COPERT 4, which does not consider the dieselgate effect) and (iv) the consideration of deterioration factors on old diesel passenger cars in HERMESv3_BU.

### 3.2.2   Non-exhaust (wear and resuspension)

HERMESv3_BU also estimates non-exhaust $PM_{10}$ and $PM_{2.5}$ traffic emissions, including road-surface, tyre and brake wear and resuspension. Emissions derived from processes of abrasion are estimated following Eq. (6):

$$Ewear_{l,i}(h) = \sum_{v=1}^{n} AADT_{v,l} * L_l * EFwear_{v,i} * S(V(h)_l) * FM(m)_l * FW(d)_l * FH(h)_l \qquad (6)$$





Where $Ewear_{l,i}(h)$ are the hourly emissions of pollutant $i$ at road link $l$ and hour $h$ [g h$^{-1}$]; $AADT_{v,l}$ is the annual average daily traffic for vehicle category $v$ at road link $l$ [n day$^{-1}$]; $L_l$ is the length of the road link $l$ [km]; $EFwear_{v,i}$ is the emission factor linked to vehicle category $v$ and pollutant $i$ [g km$^{-1}$]; $S(V(h)_l)$ is the correction factor [0.902:1.39] estimated as a function of the hourly mean vehicle travelling speed $(V(h)_l$ at road link $l$ and hour $h$ [km h$^{-1}$]; $FM(m)_l$ is the monthly factor associated to month $m$ and road link $l$ [0:12]; $FW(d)_l$ is the weekly factor associated to day $d$ and link $l$ [0:1] and $FH(h)_l$ is the hourly factor associated to hour $h$ and d link $l$ [0:1]. Both the emission and correction factors are derived from the Tier 2 methodology proposed by EMEP/EEA (2016) (Chapter 1.A.3.b.vi, Tables 3-4, 3-6 and 3-8). The number of vehicle categories is $n$.

In the case of resuspension, emissions are estimated as follows (Eq. 7):

$$Eresus_{l,i}(h) = \sum_{v=1}^{n} AADT_{v,l} * L_l * EFresus_{v,i} * S(Hrain(h)_l) * FM(m)_l * FW(d)_l * FH(h)_l \qquad (7)$$

All the parameters used in the expression are the same as the ones defined in Eq. (6) except for the resuspension emission factor linked to vehicle category $v$ and pollutant $i$ [g km$^{-1}$] ($EFresus_{v,i}$) and the correction factor $S(Hrain(h)_l)$ [0:1]. The resuspension emission factors proposed by default are vehicle type dependent (i.e. motorcycles, passenger cars, light duty vehicles, heavy duty vehicles) and derived from a measurement campaign performed in Barcelona (Amato et al., 2012a). The correction factor is estimated as a function of the number of hours after a precipitation event at road link $l$ and hour $h$ ($Hrain(h)_l$), following the expression reported by Amato et al. (2012b) (Eq. 8):

$$S(Hrain(h)_l) = (1 - e^{-r * Hrain(h)_l}) \qquad (8)$$

The formula, which is based on measurements undertaken in Barcelona (Spain) and Utrecht (The Netherlands), indicates that after a rainfall (when the mobility particles drops to values close to zero), the loading of mobile road dust mobility increases exponentially tending to reach again the maximum emission strength. The equation depends on a recovery rate ($r$) that varies according to the traffic characteristics and local climatic conditions. By default, HERMESv3_BU uses the recovery rate value derived from the Barcelona measurements, but the user can change it to other values if desired. The effect of precipitation on resuspension emissions is only applied when at least a 0.254mm h$^{-1}$ rainfall occurs (US EPA, 2011).

### 3.2.3 Gasoline evaporation

NMVOC evaporative diurnal emissions are considered in HERMESv3_BU as follows (Eq. 9):





$$E_i(x,h) = \sum_{v=1}^{n} N(x)_v * EF(T(x,d))_v * FH(T(x,h)) \qquad (9)$$

Where $E_i(x,h)$ are the hourly emissions of NMVOC at the destination grid cell $x$ and hour $h$ [g h$^{-1}$]; $N(x)_v$ is the number of registered vehicles of category $v$ in the destination grid cell $x$ [nº vehicles]; $EF(T(x,d))_v$ is the emission factor for vehicle category $v$ [g nª vehicles$^{-1}$] as a function of the daily mean outdoor temperature $T(x,d)$ for day $d$ and destination grid cell $x$ [ºC] and $FH(T(x,h))$ is the hourly factor associated to hour $h$ as a function of the hourly mean outdoor temperature $T(x,h)$ for hour $h$ and destination grid cell $x$. The number of vehicle categories is $n$.

The gridded number of registered vehicles ($N(x)_v$) are obtained combining the GHSL gridded population map with information provided by the user on registered gasoline vehicles at NUTS level 3, following the spatial operations showed in section 2.4. Contrary to the exhaust and wear emissions, evaporative emissions are considered as an area source and directly computed at the grid cell level.

Emission factors are derived from the Tier 2 method of EMEP/EEA (2016) (Chapter 1.A.3.b.v, Tables 3-5 and 3-6). Neither running loss nor hot-soak emissions are currently considered in HERMESv3_BU. This is due to the fact that these emissions mainly occur in gasoline vehicles with carburettors, and the fraction of European passenger cars and light duty vehicles post-EURO 1 with this technology is almost zero.

### 3.3 Agriculture

### 3.3.1 Fertilizers application

Hourly and spatially disaggregated NH$_3$ emissions from agricultural fertilizers are estimated following the expression reported by Paulot et al. (2014) (Eq. 10):

$$E(x,h) = \sum_{c=1}^{n} A(x)_c * C_c * \Gamma(x)_c * EF(x)_c * FD(x,d)_c * FH(h) \qquad (10)$$

Where $E(x,h)$ are the hourly NH$_3$ emissions at destination grid cell $x$ and hour $h$ [g h$^{-1}$]; $A(x)_c$ is the annual cultivated area of crop $c$ at destination grid cell $x$ [ha year$^{-1}$]; $C_c$ is the ration of cultivated to fertilised area for crop $c$ [0:1]; $\Gamma(x)_c$ is the fertilizer application rate for crop $c$ at destination grid cell $x$ [kg N ha$^{-1}$]; $EF(x)_c$ is the emission factor for crop $c$ at destination grid cell $x$ [g NH$_3$ kg N$^{-1}$]; $FD(x,d)_c$ is the daily factor for crop $c$ at destination grid cell $x$ and day $d$ [0:1] and $FH(h)$ is the hourly factor associated to hour $h$ [0:1]. The number of crop categories is $n$.





The distribution of the cultivated crop areas onto the destination grid cells $(A_c(x))$ is performed using the spatial operation capabilities of HERMESv3_BU (see section 2.4). The model combines the land use Geotiff raster reported by the CORINE Land Cover (CLC) inventory 2018 version 18 at 250x250m (CLMS, 2018) with cultivated crop area statistics at NUTS level 2 provided by the user. HERMESv3_BU performs a mapping between the different CLC and crop categories (Table A1) in

5    order to spatially distribute the statistics across the space. One limitation of this approach is that the number of agricultural land use categories in CLC is limited and therefore certain crops are assigned to the same CLC category (e.g. maize, barley, wheat, oat and rye crop categories are all mapped to the "Permanently irrigated land" CLC category). Future works will include exploring the use of more detailed datasets such as the crop type map product included in the Sentinel-2 for Agriculture portfolio (http://www.esa-sen2agri.org), which provides maps of the main crop types at 10meters resolution based on Sentinel-

10    2 and Landsat-8 imagery.

The emission factors $(EF(x)_c)$ are calculated following the methodology proposed by Bouwman and Boumans (2002), which determines that the NH$_3$ volatilisation is driven by soil pH and cation exchange capacity (CEC), type of fertilizer used (e.g. urea, ammonium, ammonium sulphate, manure), type of crop (i.e. upland, flooded) and application mode (i.e. broadcast or

15    injection). HERMESv3_BU takes the soil parameters from the International Soil Reference and Information Centre (ISRIC) World Soil Information database (Hengl et al., 2017), which reports global pH soil and CEC maps at 250m resolution. Original maps are remapped onto the user-defined grid applying a spatial intersection operation. All crops are assumed to by upland except for rice, and the application mode is assumed to be broadcast in all cases, following Paulot et al. (2014). The input on the type of fertilizer used can be distinguished by crop type and NUTS level 2.

The daily factors $(FD(x,d)_c)$ are estimated following the dynamical ammonia emission parametrization reported Gyldenkærne (2005) and Skjøth et al. (2004 and 2011), which is dependent on the outdoor air temperature, wind speed and timing of the fertilizer application, the last parameter being described with a Gauss function (Eq. 11):

$$FD(x,d)_c = e^{0.0223*T(x,d)+0.0419*WS(x,d)} * \sum_{a=1}^{3} \frac{\beta_{a,c}}{\sigma_{c,a}*\sqrt{2*\pi}} * e^{\left(\frac{(d-\tau_{c,a})^2}{-2*\sigma_{c,a}^2}\right)} \tag{11}$$

Where $T(x,d)$ is the 2 meter outdoor temperature at destination grid cell $x$ and day $d$ [ºC]; $WS(x,d)$ is the 10 meter wind speed at destination grid cell $x$ and day $d$ [m s$^{-1}$]; $\beta_{a,c}$ is the fraction of fertilizer applied to crop $c$ at stage $a$ (1: planting, 2: at growth, 3: after harvest); $\tau_{c,a}$ is the optimal application date for crop $c$ at stage $a$ [Julian day, 1:365/366]; $\sigma_{c,a}$ is the deviation around date $\tau_{c,a}$ [number of days] and $d$ is the day of the year [Julian day, 1:365/366].





Concerning the hourly distribution of emissions, the fixed temporal profile for the agriculture sector reported by Denier van der Gon et al. (2011) is proposed by default.

Figure 6.a shows the results for the Spanish total annual NH$_3$ fertilizer emissions calculated on a lambert conformal conic grid
of 4km by 4km resolution. Cultivated crop area statistics were obtained from MAPA (2017a), and information on the fertilizer application rate and type of fertilizer used by crop type were derived from both MAPA (2011 and 2017b) and Mueller et al. (2012). The time series of daily NH$_3$ emissions for the region of Aragon-Catalonia is plotted in Fig. 6.b. To calculate the daily distribution, the fraction of fertilizers applied to each crop and stage are obtained from Paulot et al. (2014), and the application dates and deviation values are derived from multiple sources, including Gyldenkærne et al., (2005), Sacks et al. (2010), and
Skjøth et al. (2011). In the particular case of barley, rye, wheat, oats and maize, the growth application dates are determined using the accumulated growing degree days (GDD) since planting (McMaster and Wilhelm, 1997). Meteorological parameters were derived from the ERA5 dataset (C3S, 2017). As shown in the example, HERMESv3_BU is able to discriminate the emission estimation by crop type, which allows quantifying the contribution of each source to the total NH$_3$. The annual emissions estimated for this Spanish region, which is considered to be one of the Europe's main NH$_3$ hotspots, are compared
against the emission fluxes derived from Infrared Atmospheric Sounding Interferometer (IASI) satellite observations (Van Damme et al., 2018) (Fig. 6.d). It is observed that results are in agreement, the emissions reported by HERMESv3_BU being just -6% lower to the ones derived from the IASI instrument. It is important to note that for this comparison, livestock emissions estimated by HERMESv3_BU have also been considered (see section 3.3.2).

### 3.3.2   Livestock

Hourly gridded NH$_3$ emissions derived from manure management activities are estimated according to the expression reported by Paulot et al. (2014) (Eq. 12):

$$E(x,h) = \sum_{a=1}^{a=4} \sum_{l=1}^{l=n} D(x)_l * \Gamma_l * \beta_l * \gamma_{a,l} * EF_{a,l} * FD(x,d)_{a,l} * FH(h) \qquad (12)$$

Where $E(x,h)$ are the hourly NH$_3$ emissions at destination grid cell $x$ and hour $h$ [g h$^{-1}$]; $D(x)_l$ is the animal density for
livestock category $l$ at destination grid cell $x$ [nº of heads cell$^{-1}$]; $\Gamma_l$ is the Nitrogen (N) excretion rate for livestock $l$ [g N head$^{-1}$]; $\beta_l$ is the fraction of total ammoniacal nitrogen (TAN) content of the excreta from livestock $l$ [0:1]; $\gamma_{p,l}$ is the fraction of total excreta associated to activity $a$ (1: housing, 2: yarding, 3: storage, 4: grazing) for livestock $l$ [0:1]; $EF_{a,l}$ is the emission factor for livestock $l$ and activity $a$ [g NH$_3$ g N$^{-1}$]; $FD(x,d)_{a,l}$ is the daily factor for livestock $l$ and activity $a$ at destination grid cell $x$ and day $d$ [0:1] and $FH(h)$ is the hourly factor associated to hour $h$ [0:1]. The number of livestock categories is $n$.


The estimation methodology and emission factors follow the Tier 2 approach proposed by EMEP/EEA (2016) (Chapter 3.B, Table 3.9), which uses a mass-flow approach based on the concept of a flow of TAN through the manure management system.

Regarding the animal density data ($D(x)_l$), HERMESv3_BU uses as a basis the gridded livestock population from the Gridded Livestock of the World version 3 (GLWv3, Gilbert et al., 2018), which provides raster maps of global population densities of cattle, buffaloes, horses, sheep, goats, pigs, chickens and ducks for 2010 at a spatial resolution of 0.083333 degrees. HERMESv3_BU adjusts the original data to match province-level official records from most recent years. These official statistics, which need to be provided by the user, are also used to distribute each general livestock GLWv3 group (e.g. pigs) into specific categories (e.g. fattening pigs between 50 and 80kg, boars, sows not yet covered). This disaggregation is relevant due to the different types of feeding received by each animal type, which subsequently affect the levels of N and TAN content in their excreta. The remapping and adjustment of the GLWv3 original data to the destination gridded domain is performed using the spatial operation tools described in section 2.4. HERMESv3_BU currently considers a total of 36 livestock categories, which are grouped into five main groups: pigs (10), cattle (11), poultry (2), goats (6) and sheep (7). Other livestock groups (i.e. buffaloes, horses and ducks) are not currently included due to their low contribution to NH₃ emissions in Europe.

Daily factors ($FD(x,d)_{a,l}$) are estimated following the dynamical emission parametrization reported in Gyldenkærne (2005) and Skjøth et al. (2004 and 2011). For housing operations, the daily factors are assumed to be dependent on the barn air temperature and ventilation rate, following Eq. (13):

$$FD(x,d)_{housing,l} = (T(x,d)_{housing,l}^{0.89} + V(x,d)_{housing,l}^{0.26}) \tag{13}$$

Where $T(x,d)_{housing,l}$ is the barn temperature associated to the housing of livestock category $l$ at destination grid cell $x$ and day $d$ [ºC] and $V(x,d)_{housing,l}$ is the ventilation rate associated to the housing of livestock category $l$ at destination grid cell $x$ and day $d$ [m s⁻¹]. Both parameters are calculated as a function of the outdoor 2 m temperature and 10m wind speed considering the parametrizations reported in Gyldenkærne (2005), which take into account if the livestock are kept in open or closed barns. HERMESv3_BU assumes that pigs and poultry are kept in closed barns, while cattle, sheep and goats are kept in open barns, following Backes et al. (2016). The daily factors for yarding and storage activities also follow Eq. (13), but using wind speed and air temperature. Finally, for grazing activities the temporal variability is linked to the availability of grass and to its growing period. Therefore, the daily distribution are estimated using Eq. (11) and considering only the growing stage of grass.

Concerning the hourly distribution of emissions, the same profile proposed for fertilizer application is proposed.




Figure 6.c shows the results of the Spanish annual NH₃ livestock emissions calculated on a lambert conformal conic grid of 4kmx4km resolution. Animal number statistics at NUTS level 3 were obtained from MAPA (2017c), and information on the N excretion rate for each livestock category was derived from MAPA (2017b). The TAN content data was obtained from Antezana et al. (2016) for pig categories and EMEP/EEA (2016) (Chapter 3.B, Table 3.9) for the rest of animals. The time series of daily NH₃ emissions for the region of Murcia is plotted in Fig. 6.d. Meteorology is derived from ERA5. As shown in the example, HERMESv3_BU is able to discriminate the emission estimation by livestock group, which allows quantifying the contribution of each source to the total NH₃. The annual emissions estimated for this Spanish region, also considered a major European NH₃ hotspot, are again compared against the emission fluxes derived from IASI (Van Damme et al., 2018) (Figure 6.e). The HERMESv3_BU results include both livestock and fertilizers emissions. It is observed that results are in the same order of magnitude, HERMESv3_BU reporting 21% less emissions.

### 3.3.3 Crop operations

Particulate matter emissions released during soil cultivation and crop harvesting activities are estimated following Eq. (14):

$$E(x,h) = \sum_{o=1}^{m}\sum_{c=1}^{n} A_c(x) * EF_{c,o} * FM(m)_{c,o} * FW(d) * FH(h) \tag{14}$$

Where $E(x,h)$ are the hourly PM$_{10}$/PM$_{2.5}$ emissions at destination grid cell $x$ and hour $h$ [g h$^{-1}$]; $A_c(x)$ is the annual cultivated area of crop $c$ at destination grid cell $x$ [ha year$^{-1}$]; $EF_{c,o}$ is the emission factor for crop $c$ and operation $o$ [g PM$_{10}$/PM$_{2.5}$ ha$^{-1}$]; $FM(m)_{c,o}$ is the monthly factor associated to crop $c$, operation $o$ and month $m$ [0:1]; $FW(d)$ is the weekly factor associated to day $d$ [0:1] and $FH(h)$ is the hourly factor associated to hour $h$ [0:1]. The number of crop categories is $n$.

The model uses the emission factors reported by the EMEP/EEA (2016) Tier 2 methodology (Chapter 3.D, tables 3.6 and 3.8). The methodology takes into account emissions happening in four different types of crops (i.e. wheat, rye, barley and oat). Gridded crop areas are estimated using the same approach described in the fertilizers sector (see section 3.3.1). Considering the monthly distribution of emissions, specific weight factors are applied based on the soil cultivation and harvesting calendars reported by Sacks et al. (2010). The daily and hourly profiles used by default are the ones recommended by EMEP/EEA (2016) for temporally allocating agricultural machinery activities (Chapter 1.A.4.c ii, table 5.1).

### 3.4 Residential and commercial combustion

Emissions from residential and commercial small combustion plants are estimated as follows (Eq. 15):

$$E_i(x,h) = \sum_{f=1}^{n} FC_f(x) * EF_{f,i} * FD(x,d)_f * FH(h)_f \tag{15}$$





Where $E_i(x, h)$ are the hourly emissions of pollutant $i$ and hour $h$ [g h$^{-1}$]; $FC_f(x)$ is the annual gridded fuel consumption data for fuel category $f$ and destination grid cell $x$ [GJ year$^{-1}$]; $EF_{f,i}$ is the emission factor linked to the consumption of fuel $f$ and pollutant $i$ [g GJ$^{-1}$]; $FD(d, x)_f$ is the gridded daily factor associated to fuel type $f$, destination grid cell $x$ and day $d$ [0:1] and $FH(h)_f$ is the hourly factor associated to fuel type $f$ and hour $h$ [0:1]. The number of fuel categories is $n$.

The gridded consumption data is calculated combining the fuel statistic consumptions at NUTS level 3 provided by the user with the GHSL population data, as previously described in section 2.4. HERMESv3_BU considers the following fuel types: natural gas, liquefied petroleum gas (LPG), heating diesel oil, wood and coal. For all of them, emission factors are obtained from the EMEP/EEA (2016) tier 2 approach (chapter 1.A.4, tables 3-10, 3-18, 3-19, 3-21, 3-31 and 3-37). In the particular case of wood, the proposed emission factors are obtained as an average of the values reported for the different types of appliances (i.e. fireplace, conventional stove, conventional boiler, ecolabelled boiler and pellet stove, tables 3-14, 3-17, 3-18, 3-24 and 3-25). The average emission factors are obtained considering the appliances shares reported by Denier van der Gon et al. (2015). It is also important to note that wood-related PM$_{10}$ and PM$_{2.5}$ emission factors take into account the condensable fraction of PM.

The temporal distribution of annual emissions is performed using gridded daily temporal profiles, which are derived according to the heating Degree Day (HDD) concept. The HDD is an indicator used as a proxy variable to reflect the daily energy demand for heating a building (Quayle and Diaz, 1980). The original expression is reformulated according to Mues et al., (2014) to consider those combustion processes that are not only related to space heating but also to other activities that remain constant throughout the year (i.e. water heating, cooking) (Eq. 16):

$$FD_f(x, d) = \frac{HDD(x, d) + f_f * \overline{HDD}(x)}{(1 + f_f) * \overline{HDD}(x)} \tag{16}$$

Where $HDD(x, d)$ is the heating degree day factor for grid cell $x$ and day of the year $d$ [0:1]; $\overline{HDD}(x)$ is the yearly average of the heating degree day factor per grid cell $x$ and $f_f$ is a constant offset that indicates the share of the fuel $f$ that is used for activities not related to space heating [0:1]. By default, $f_f$ is considered to be 0 for wood and 0.2 for the other fuels, following the European household energy statistics reported by Eurostat (2018). $HDD(x, d)$ and $\overline{HDD}(x)$ are estimated as shown in Eq. (17) and (18) (Quayle and Diaz, 1980):

$$HDD(x, d) = \max(T_b - T_{2m}(x, d), 1) \tag{17}$$




$$\overline{HDD}(x) = \frac{\sum_1^N HDD(x,d)}{N} \tag{18}$$

Where $T_b$ is the threshold temperature above which a building needs no heating (i.e. heating appliances will be switched off) [ºC], $T_{2m}(x,d)$ is the daily mean 2m outdoor temperature for grid cell $x$ and day $d$ [ºC] and $N$ is the number of days of the simulation year [365 or 366]. Following Spinoni et al. (2015), who developed gridded European degree-day climatologies, we assume $T_b = 15.5°C$, a value also suggested by the UK MET-Office. The $HDD(x,d)$ value increases with increasing difference between the outdoor and base temperatures. Note that a minimum value of 1 is assumed instead of 0 to avoid numerical problems.

Two profiles are proposed in HERMESv3_BU for the hourly distribution of emissions. The first one applies only to residential wood burning emissions and it is a combination of existing profiles derived from citizen interviews performed in Norway and Finland (Finstad et al., 2004 and Gröndahl et al., 2010) as well as from long-term measurements of the wood burning fraction of black carbon in Athens (Athanasopoulou et al., 2017). The second profile (applicable to all other fuel types) is equivalent to the one proposed by Denier van der Gon et al. (2011) for residential combustion emissions. The wood-related profile presents an intense peak in the evening hours, but not during the morning (Grythe et al., 2019). This fact is related to a common practice in Europe of using fireplaces and other types of wood-burning appliances mainly in the evening.

### 3.5 Other mobile sources

### 3.5.1 Shipping in port areas

Hourly emissions related to fuel combustion processes occurring in main and auxiliary maritime engines during manoeuvring and hoteling operations in port areas are estimated as follows (Eq. 19):

$$E_{p,f,i}(x,h) = \sum_{v=1}^{n}\sum_{e=1}^{2} N_{p,v} * S(x)_{p,f} * t_{p,v,f} * P_{v,e} * LF_{v,e,f} * EF_{v,e,f,i} * FM(m)_{v,p} * FW(d)_p * FH(h)_{v,p} \tag{19}$$

Where $E_{p,f,i}(x,h)$ are the hourly emissions of pollutant $i$ at port $p$, destination grid cell $x$ and hour $h$ during phase $f$ (i.e. manoeuvring, hoteling) [g h⁻¹]; $N_{p,v,f}(x)$ is the number of annual operations associated to vessel $v$ (i.e. liquid bulk ship, dry bulk carrier, general cargo, ro-ro, cruiser, ferry, container, tug or others) at port $p$ [operation year⁻¹]; $S(x)_{p,f}$ is the spatial weight factor associated to port $p$ and destination grid cell $x$ during phase $f$ [0:1]; $t_{p,v,f}$ is the time spent by vessel $v$ to complete phase $f$ at port $p$ [h]; $P_{v,e}$ is the average power of vessel's $v$ engine $e$ (1: main, 2: auxiliary) [kW]; $LF_{v,e,f}$ is the average load factor of vessel's $v$ engine $e$ of during phase $f$ [0:1]; $EF_{v,e,f,i}$ is the emission factor for pollutant $i$ of vessel's $v$ engine $e$ during phase $f$ [g kWh⁻¹]; $FM(m)_{v,p}$ is the monthly factor associated to month $m$, vessel $v$ and port $p$ [0:1]; $FW(d)_p$ is the weekly


factor associated to day of the week $d$ and port $p$ [0:1] and $FH(h)_p$ is the hourly factor associated to hour $h$ and port $p$ [0:1]. The number of vessel categories is $n$.

The estimation methodology and corresponding emission factors are derived from the EMEP/EEA (2016) Tier 3 approach
(Chapter 1.A.3.d, table 3-10). Information on the vessel's technical characteristics are obtained from Trozzi (2010), including: (i) the average power of vessel's engines (as a function of the gross tonnage) and (ii) the engine type (i.e. slow speed diesel, medium speed diesel, high speed diesel, gas turbine, steam turbine) and fuel class (i.e. Marine diesel oil, marine gas oil and bunker fuel oil) assigned to each vessel category. The values of engine load factors and operation times per phase and vessel are obtained from Entec (2002). The remaining information, which is port-dependent, needs to be provided by the user.

The gridded weight factors ($S(x)_{p,f}$), which are used to spatially allocate the emissions, are calculated by the model through a spatial intersection between the destination domain and two shapefiles representing the areas of manoeuvring and hoteling of each port. Both shapefiles need to be provided by the user. Figure 7.a shows an example of the total $NO_x$ annual emissions [t year$^{-1}$] estimated for the Port of Barcelona on a 1kmx1km regular grid. Both the manoeuvring and hoteling shapefiles used for
the spatial distribution were digitalised using as a basis an infrastructure map provided by the Barcelona's Port Authority (APB, personal communication). The hoteling layer (in red) consists on a multipolygon shapefile, each polygon representing one of the docks in which the ships operate. A specific weight was assigned to each dock as a function of its usage, allowing a more realistic distribution of the total emissions. On the other hand, manoeuvring emissions were distributed homogeneously between all the cells intersected by the corresponding shapefile (light blue layer). Annual emissions are compared against the
results reported by the APB 2013 inventory (APB, 2016) (Fig. 7.b). Results show that HERMESv3_BU estimates lower emissions both for $NO_x$ and $PM_{10}$. The difference can be related to the different years of reference and subsequently number of vessels operations considered, as well as to the fact that the APB inventory assumes a main engine load factor of 20% during hoteling operations, while in HERMESv3_BU the load factor is null following the recommendations in Entec (2002).

**3.5.2    Recreational boats**

Emissions derived from pleasure boat activities can be estimated in HERMESv3_BU following Eq. (20):

$$E_i(x,h) = \sum_{b=1}^{n} N_b(x) * T_b * P_b * LF_b * EF_{b,i} * FM(m) * FW(d) * FH(h) \qquad (20)$$

Where $E_i(x,h)$ are the hourly recreational boat emissions of pollutant $i$ at destination grid cell $x$ and hour $h$ [g h$^{-1}$]; $N_b(x)$ is
the number of pleasure boats associated to category $b$ and destination grid cell $x$ [nº boats]; $T_b$ is the number of hours that the pleasure boat of category $b$ is used during one year [h]; $P_b$ is the engine nominal power associated to the pleasure boat of





category $b$ [kW]; $LF_b$ is the engine load factor associated to the pleasure boat of category $b$ [0:1]; $EF_{b,i}$ is the emission factor linked to pleasure boat of category $b$ and pollutant $i$ [g kWh$^{-1}$]; $FM(m)$ is the monthly factor associated to month $m$ [0:1]; $FW(d)$ is the weekly factor associated to day of the week $d$ [0:1] and $FH(h)$ is the hourly factor associated to hour $h$ [0:1]. The number of recreational boat categories is $n$.

The parameters $EF_{b,i}$, $LF_b$, $T_b$ and $P_b$ are derived from the EMEP/EEA (2016) Tier 3 approach (Chapter 1.A.5.b, Table 3-11). The gridded number of pleasure boats ($N_b(x)$) is computed by combining official statistics of registered recreational crafts with a raster file that simulates the distribution of recreational boat activities along the coast. Both datasets need to be provided by the user. Figure 7.c shows the spatial distribution of Spanish recreational boats along the Costa Brava region (coast of

Catalonia). The spatial proxy is based on the location of each Spanish marina and associated number of docks, which are also represented in the map (Fondear, 2019). A raster interpolation was performed to simulate the activities of recreational boats nearby the marinas, considering the number of docks as a weight and assuming that no operations are happening beyond the territorial waters (i.e. 12 nautical miles away from the coastline). A hot spot region is observed near Cap de Creus (headland located at the far northeast of Catalonia) due to the presence of the largest Spanish marina (Empuriabrava, with 5,000 docks).

Total number of recreational boats per type of boat were derived from ICOMIA (2016). Figure 7.d shows the total annual emissions estimated with HERMESv3_BU for Spain. To the author's knowledge, no other national emission inventory is currently available for this pollutant source. Emissions are contrasted against an inventory of pleasure boats estimated for the Baltic Sea (Johansson et al., 2017) to assess that the results are within the same range of magnitude. HERMESv3_BU reports higher NO$_x$ (4.1 times) and lower CO and NMVOC (0.7 and 0.5 times, respectively) emissions. This is probably due to the

different fleet characteristics of each region. While in Spain more than 40% of the boats are related to large diesel motor sail boats, in the Baltic Sea region this category only accounts for less than 15% of the total fleet.

### 3.5.3 Agricultural machinery

Emissions related to the use of agricultural equipment (i.e. two-wheel tractors, agricultural tractors and harvesters) are

estimated following Eq. (21):

$$E_i(x,h) = \sum_{e=1}^{n} N_e(x) * T_e * P_e * LF_e * EF_{e,i}(P_e) * (1 + DF_{e,i}) * FM_e(m) * FW(d) * FH(h) \qquad (21)$$

Where $E_i(x,h)$ are the hourly emissions of pollutant $i$ at destination grid cell $x$ and hour $h$ [g h$^{-1}$]; $N_e(x)$ is the number of agricultural equipment associated to category $e$ and destination grid cell $x$ [n° equipment]; $T_e$ is the number of hours that the

equipment of category $e$ is used during one year [h]; $P_e$ is the engine nominal power associated to the equipment of category $e$



[kW]; $LF_e$ is the engine load factor associated to the equipment of category $e$ [0:1]; $EF_{e,i}(P_e)$ is the emission factor linked to the agricultural equipment of category $e$ and pollutant $i$ [g kWh$^{-1}$] as a function of the engine nominal power $P_e$; $DF_{e,i}$ is the deterioration adjustment factor for the equipment of category $e$ and pollutant $i$; $FM_e(m)$ is the monthly factor associated to month $m$ and equipment category $e$ [0:1]; $FW(d)$ is the weekly factor associated to day $d$ [0:1] and $FH(h)$ is the hourly factor associated to hour $h$ $l$ [0:1]. The number of agricultural equipment categories is $n$.

HERMESv3_BU takes as an input the total number of agricultural equipment and corresponding engine nominal power registered at the NUTS 3 level. The spatial allocation of this data onto the destination domain is performed considering the CLC non-irrigated arable land category as a proxy and applying the spatial operations described in section 2.4. Both the emission and deterioration adjustment factors are derived from the EMEP/EEA (2016) Tier 3 methodology (chapter 1.A.4, table 3-6 and table 3-11). The load factor adjustments proposed by default are the ones reported by Winther and Nielsen (2006) (i.e. 0.4 for two-wheel tractors, 0.5 for agricultural tractors and 0.8 for harvesters). Emissions from two-wheel tractors and agricultural tractors are disaggregated at the monthly level considering the crop calendars associated to the cultivation operation. In the case of the harvester emissions, the monthly factors are associated to the harvesting period of the non-irrigated arable crops (Sacks et al., 2010).

### 3.5.4 Landing and take-off cycles in airports

Hourly aircraft landing and take-off (LTO) emissions occurring at airports are estimated according to Eq. (22):

$$E_{a,f,i}(x,h) = \sum_{p=1}^{n} N(m)_{a,p,f} * S(x)_{a,f} * EF_{p,f,i} * FW(d)_a * FH(h)_{a,f} \tag{22}$$

Where $E_{a,f,i}(x,h)$ are the hourly emissions of pollutant $i$ at airport $a$, destination grid cell $x$ and hour $h$ during phase $f$ (i.e. approach, landing, taxi-in, post taxi-in, pre taxi-out, taxi-out, take-off, and climb-out) of the LTO cycle [g h$^{-1}$]; $N(m)_{a,p,f}$ is the number of monthly operations associated to aircraft $p$ at airport $a$ during phase $f$ for month $m$ [op month$^{-1}$]; $S(x)_{a,f}$ is the spatial weight factor associated to airport $a$ and destination grid cell $x$ during phase $f$ [0:1]; $EF_{p,f,i}$ is the emission factor for pollutant $i$ associated to aircraft $p$ and phase $f$ [g op$^{-1}$]; $FW(d)_a$ is the weekly factor associated to day $d$ and airport $a$ [0:1] and $FH(h)_{a,f}$ is the hourly factor associated to hour $h$ phase $f$ and airport $a$ [0:1].

Depending on the LTO phase, different emission processes and subsequently emission factors are considered. For taxi-in, taxi-out, take-off, climb out and approach operations the emission factors correspond to the fuel combustion in the main engines ($EFmain_{p,f,i}$) (Eq. 23):





$$EFmain_{p,f,i} = E_p * t_{a,p,f} * EFengine_{p,f,i} \qquad (23)$$

Where $E_p$ is the number of engines associated to aircraft $p$ [engine]; $t_{a,p,f}$ is the time spent by aircraft $p$ to complete phase $f$ at airport $a$ [s] and $EFengine_{p,f,i}$ is the emission factor for pollutant $i$ of the main engine associated to aircraft $p$ and phase $f$ [g s$^{-1}$·engine$^{-1}$ op$^{-1}$]. The number of engines and emission factors associated to each aircraft category are derived from the

EMEP/EEA (2016) Tier 3 approach (Annex 5 spreadsheets to the chapter 1.A.3.a). For taxi-in and taxi-out, times are also obtained from the same source, whereas for the other operations (take-off, climb out and approach) different values are assumed as a function of the type of aircraft (i.e. wide-body planes, narrow-body planes, business planes and light planes with piston engines) (Dellaert and Hulskotte, 2017).

For landing operations, particulate matter emission factors ($EFwear_{p,i}$) related to the wear of the aircraft brakes and tyres are considered (Eq. 24) (Morris, 2006):

$$EFwear_{p,i} = MTOW_p * EFwear_i \qquad (24)$$

Where $MTOW_p$ is the maximum take-off weight associated to aircraft $p$ [tonne] and $EFwear_i$ is the emission factor for

pollutant $i$ [g tonne$^{-1}$ op$^{-1}$], which is taken from Morris (2006).

Finally, emissions during the pre taxi-out and post taxi-in operations are linked to the fuel combustion in the auxiliary power units (APU). The corresponding emission factors ($EFaux_{p,i}$) are estimated as follows (Eq. 25) (Watterson et al., 2004):

$$EFaux_{p,i} = t\_apu_{a,p,f} * EFapu_{p,i,f} \qquad (25)$$

Where $t\_apu_{a,p,f}$ is the APU running time associated to aircraft $p$, phase $f$ and airport $a$ [s] and $EFapu_{p,i}$ is the emission factor for pollutant $i$ of the APU engine associated to aircraft $p$ and phase $f$ [g s$^{-1}$·op$^{-1}$]. Data on the type of APU installed in each type of aircraft and corresponding emission factors are obtained from Watterson et al. (2004).

The spatial weight factors ($S(x)_{a,f}$) to spatially distribute the estimated emissions also varies according to the LTO cycle's phase. Taxi-in, post taxi-in, pre taxi-out and taxi-out emissions are assigned at the ground level; taking into account digitised airport areas. Emissions are mapped to the grid cells that spatially intersect with the airport polygons, which need to be provided by the user in a shapefile. Take-off and landing emissions are also allocated at the ground level, but they are horizontally distributed across the grid cells that intersect with the digitised runways (also provided by the user in a shapefile). The user

has the option to define specific weights to each runway as a function of its usage (for departures, arrivals or both). Finally,





emissions from approach and climb-out operations are allocated on a 3D basis, considering the trajectory outlined between the origin/end of the runway and 1000 m of altitude. This operation is performed in HERMESv3_BU using as input a shapefile representing the air trajectories and information of the approach and take-off angles for each runway.

### 3.6 Speciation

This process disaggregates the calculated primary pollutants (i.e. $NO_x$, NMVOC, $PM_{2.5}$) into the more detailed species defined by the chemical mechanism of interest. .Specific speciation CSV files, which contain a set of profiles with numerical factors for converting the primary pollutants (e.g. $NO_x$) to output model species (e.g. NO, $NO_2$ and HONO), are assigned to each pollutant sector. The number of speciation profiles considered varies according to the pollutant sector of interest. For instance, in the case of residential combustion the number of speciation profiles proposed is equal to the number of fuel types considered

(e.g. natural gas, biomass), whereas in the case of road transport specific profiles are assigned to each vehicle category. The assignment between profiles and pollutant categories are performed following the cross-referencing system described in section 2.3.

The speciation factors are mass-based (i.e. g of chemical specie · g of source pollutant$^{-1}$) for $NO_x$ and $PM_{2.5}$. For $NO_x$ emissions

are divided into 90% NO and 10% $NO_2$ for all sectors (Houyoux et al., 2000) except for road transport, where vehicle-dependent speciation factors are derived from EMEP/EEA (2016) (chapter 1.A.3.b.i-iv, table 3.87) and the works by Carslaw et al. (2016) and Rappenglueck et al. (2013). For $PM_{2.5}$, the speciation factors are derived from multiple sources including EMEP/EEA (2016) and the SPECIATE (Simon et al., 2010) and SPECIEUROPE (Pernigotti et al., 2016) databases. For the rest of the calculated primary pollutants (i.e. CO, $NH_3$, $SO_2$, $PM_{10}$) a default speciation factor of 1 is proposed for all sources.

All computed gas-phase species are converted from mass to moles using a molecular weight CSV file that is included in the model database (built-in data file).

In the case of NMVOC, the speciation factors used by HERMESv3_BU are mol-based (mol of chemical specie · g of source pollutant$^{-1}$) and are estimated following the expression proposed by Li et al. (2014) (Eq. 26):

$$SF_{\bar{e},p} = \sum_{j=1}^{n} \frac{X_{j,p}}{MW_j} * C_{j,\bar{e}} \qquad (26)$$

Where $X_{j,p}$ is the mass fraction of chemical compound $j$ to total NMVOC emissions for speciation profile $p$, $MW_j$ is the molecular weight of chemical compound $j$ and $C_{j,\bar{e}}$ is the mole-based conversion factor of chemical compound $j$ to destination chemical species $\bar{e}$. $X_{j,p}$ values are obtained from multiple sources including EMEP/EEA (2016) and the NMVOC SPECIATE database (Simon et al., 2010), and $MW_j$ and $C_{j,\bar{e}}$ are obtained from the mechanism-dependent mapping tables proposed in Carter

(2015). The number of individual chemical compounds considered in the total NMVOC is $n$.





For reference, the HERMESv3_BU test case input database currently includes speciation profiles for the Carbon Bond 05 (CB05) (Whitten et al., 2010) gas-phase mechanism, and the fifth-generation aerosol module (AERO5) (Roselle et al., 2008). Following the formats of the associated input files, the user can create its own speciation profiles for other mechanisms of interest and also using alternative sources of information.

## 4 Technical implementation

HERMESv3_BU is coded using Python 3.7.X and requires numpy (>D1.16.0), NetCDF4 (>D1.3.1) under HDF5 in parallel mode, pandas (>D0.22.0), geopandas (>D0.4.0), pyproj (>D1.9.5.1), configargparse (>D0.11.0), cf_units (>D1.1.3), timezonefinder (>D2.1.0), mpi4py (>D3.0.0), pytest (>D3.6.1), shapely (D>1.6.4), scipy (D>0.14.1) and rasterio (D>1.0.21) Python libraries.

The emission core of each pollutant sector (i.e. calculation, temporal and spatial distribution and speciation) is parallelised using one of the following strategies:

- Road link partition: This strategy is applied to the road transport sector. The original digitalised road transport network is split according to the number of processors to be used. Each processor computes the emissions of the road links subgroup that has been assigned with. The partition of the data is automatically balanced with respect to the number of processors used.

- Point source partition: This strategy is used for the point source sector. The approach is equal to the one described above, but in this case the partition is applied to the point source facilities.

- Polygon partition: This strategy is applied to the shipping and aviation sectors. Each processor is responsible for calculating the emissions of a subset of the total number of infrastructures (i.e. ports or airports) included in the working domain.

- Grid partition: This strategy is applied to all the sectors that are computed at the destination grid cell level once the original input activity data has been mapped onto the working domain (i.e. agricultural sources, residential and commercial combustion, recreational boats and gasoline evaporative emissions). The destination working domain is divided into subgroups of row-major consecutive cells. Each subgroup or chunk contains only those grid cells that have activity data information. The number of divisions is balanced and equal to the number of processors to be used.

In all cases, emissions are computed independently in each processor and for each partition (i.e. group of road links, point sources, polygons or grid cells). As a result of the emission calculation process, each individual computational processor ends up containing the gridded hourly estimated emissions for the corresponding section of the working domain. Once this status is



reached for all the processors (i.e. emissions have been estimated for all partitions and pollutant sectors), HERMESv3_BU starts the execution of the writing process. The model can perform this task in serial or in parallel mode, regardless of the processors used for the emission calculation process. If the user configures the model to run with more than one writing processor, HERMESv3_BU will decompose the destination working domain into horizontal sections (as many as processors

selected), maintaining each row undividable. Each processor will first gather the estimated emissions of the corresponding writing section and then compute the total sum per grid cell, vertical layer and time step. During this process, unit conversions also will take place. Finally, each individual section will write the emission 4D array (hour, vertical layer, longitude, latitude) onto the corresponding memory section of the NetCDF output file. When only one writing processor is used, the same process is performed but without applying the aforementioned domain partition.

The number of processors used to calculate each pollutant sector and to perform the writing function are defined by the user in the general configuration file (see section 2.2). The distribution of the defined computational resources is performed using a sector manager function under the framework of the Message-Passing Interface (MPI) protocol (mpi4py Python library). MPI creates a global communicator and the sector manager splits it into a group of sub-communicators (as many as pollutant

sectors involved). This allows HERMESv3_BU to exclusively assign each group of processors to a specific sector and therefore isolate the corresponding calculation processes.

Figure 8 shows an illustrative example of the emission calculation and writing parallel implementations. The example considers four pollutant sectors: a point-type sector (e.g. point sources), a line-type sector (e.g. road transport), a polygon-type

sector (e.g. aviation) and a grid-type sector (e.g. fertilizers). A total of 17 processors are defined for the emission calculation process (Fig. 8.a), which are split as follows: 7 for point sources, 3 for road transport, 3 for aviation and 4 for fertilizers. Each processor is assigned to an individual element (or group of elements) of the corresponding pollutant sector and performs the calculation of the gridded hourly emissions (Fig. 8.b). Those cells that are not intersected with any emission source are left without information. Once all the emission calculation processors have finished, the writing process begins, in this case using

a total of 3 processors. The working domain is accordingly split into 3 horizontal sections, the first two consisting of 4 completed rows and the last one of 2 (Fig. 8.c). For each grid cell of each writing section, the corresponding processor gathers and sums the emissions estimated for all the different sources (Fig. 8.d). The information stored in each writing processor is then written in the corresponding section of the output NetCDF file.

The supercomputer MareNostrum4, hosted by the BSC (www.bsc.es/marenostrum/marenostrum), was used to test the capability of HERMESv3_BU to scale up the emission calculation processes. HERMESv3_BU was executed using a number of cores from 1 to 256, doubling the number in each successive test. The proposed testing domain was a lambert conformal conic domain of 4km by 4 km with 397 rows, 397 columns and 15 vertical layers covering the Iberian Peninsula (Fig. 6.a). Hourly emissions for Spain were estimated for 24 time steps and for each one of the pollutant sectors described in section 3.

As shown in Fig. 9, for all sectors the computational time decreases as the number of cores is increased. Traffic is the pollutant sector that requires the largest amount of time to be completed, the total execution time decreasing from 48,585.5s to 450.3s when changing from 1 to 256 processors. Similar time reduction rates are observed for other sectors when comparing the

results obtained with 1 and 256 processors: livestock (from 17,530.2 s to 53.2 s), point sources (from 5,107.7 s to 75.4 s), residential and commercial combustion (from 3,230.7 s to 27.0 s) and crop fertilizers (from 2,814.7 s to 39.1 s). For the rest of the sectors (i.e. agricultural machinery, crop operations, recreational boats, shipping in ports), execution times are already considerably low when only using one processor (between 19.5 s and 326.5 s). Note that for aviation and shipping in port areas, the number of processors with which HERMESv3_BU can be executed is limited to the number of infrastructures

considered in the domain of study (i.e. 47 airports and 48 ports in this example, respectively). Traffic is by far the most time demanding pollutant sector due to a combination of several facts: (i) large amount of sources: the sector considers a road network with more than 100,000 road links and includes 491 vehicle categories, (ii) multiple emission processes: the sector estimates emissions derived from hot and cold exhaust gases, tyre, brake and road wear, resuspension and gasoline evaporation and (iii) meteorological-dependent functions: three of these processes are estimated using meteorological-dependent

parametrizations (i.e. cold exhaust and gasoline evaporation takes into account outdoor temperature, while resuspension considers precipitation information).

The execution of each sector is performed in parallel (not sequential). Therefore, the total execution time of HERMESv3_BU for the emission calculation process is equal to the sector's largest execution time. Following with the proposed example, if

the user wants to perform the emission calculation process in less than 8 minutes, a total of 298 processors should be used with the following distribution: 256 for traffic, 16 for point sources, 12 for livestock, 4 for residential and commercial combustion, 4 for crop fertilizers and 1 for each one of the other sectors. The computational time needed to gather and write all the estimated emissions into a NetCDF output file are very low compared to the emission estimation time (i.e. between 13 and 25 seconds depending on the number of writing processors selected, not shown).

**5    Conclusions**

This paper presents HERMESv3_BU, a stand-alone and open-source emission model that estimates high resolution spatial and hourly bottom-up anthropogenic emissions for air quality modelling. The tool combines state-of-the-art estimation approaches with local activity and emission factors along with meteorological data.

The main characteristics of HERMESv3_BU are as follows:



- Multiple map projections and model grids: The model supports emission calculations in regular and rotated lat-long, Lambert conformal conic and Mercator map projections. Users can freely define their working domain introducing information on the starting x-y coordinates, number of grid cells in each direction, and physical size of the grid cells.

- Multiple pollutant sources: The model includes bottom-up emission estimation methodologies for several anthropogenic emission sources, including point sources, road transport (hot and cold exhaust, wear and resuspension, gasoline evaporation), agriculture (i.e. fertilizers application, livestock, crop operations), residential and commercial combustion and other mobile sources (i.e. shipping in port areas, recreational boats, agricultural machinery, landing and take-off cycles in airports). Users can estimate emissions for each individual source or a combination thereof.

- Subsetting of pollutant categories and regions: For each pollutant sector, users can select individual pollutant
10 categories to be considered during the emission calculation process (e.g. 491 vehicle categories for road transport, 28 crop types for fertilizer application). Similarly, users can define a region of interest within the working domain (i.e. administrative areas or self-defined polygons) for which the emissions should be calculated. These functionalities can become useful when studying the contribution of certain categories/regions to total emissions (e.g. diesel vehicles from a selected province) or when performing source attribution modelling studies.

- Emission estimation approaches: HERMESv3_BU integrates estimation methodologies and emission factors that are mostly based on (but not limited to) the approaches reported by the European EMEP/EEA 2016 emission inventory guidebook. The model also includes meteorological-dependent functions to take into account the dynamical component of the emission processes, both in terms of spatial and temporal allocation (e.g. plume rise calculations for point sources, effect of temperature and wind speed on the volatilization of agricultural $NH_3$).

- Applicability: HERMESv3_BU is designed so that it can be applicable to any European country/region where the required input data is available. Global and regional state-of-the-art datasets are considered in the model to decrease the amount of information that needs to be provided by the user, including land use data (CORINE Land Cover inventory), livestock distribution (Gridded Livestock of the World), population maps (Global Human Settlement Layer) and soil properties (World Soil Information database). The application of the tool to non-European regions
25 could be also possible, but it would imply to review much more in detail the emission factors databases and estimation methodologies that are proposed by default in the model.

- Spatial operations: HERMESv3_BU contains a set of GIS functionalities that helps users manipulate and generate georeferenced data files related to emissions modelling (i.e. ESRI shapefiles and Geotiff rasters). The operations included in the model allow developing automatically individual spatial surrogates or remapping spatial data from
30 one gridded domain to another, among others.

- Emission outputs compatible with multiple air quality models: The output NetCDF emission files follow the convention of multiple chemical transport models, including CMAQ, WRF-CHEM and NMMB-MONARCH. In the



particular case of road transport, the resulting link-level emissions can be also used as input for the R-LINE Gaussian dispersion model.

- Parallel implementation: The parallelization of the emission calculation and writing processes allows scaling up the execution time of the model, which may be relevant when using the model for air quality forecasting applications.

Several emission outputs obtained with HERMESv3_BU are provided in this paper and compared to other existing emission datasets to illustrate its potential. Future work will focus on performing a full evaluation of the emission results obtained for Spain through comparisons with other inventories (e.g. EDGAR, EMEP) and using air quality modelling.

HERMESv3_BU currently experiences some limitations. The first one is the non-inclusion of a submodule to estimate NMVOC emissions related to the use of solvents. For this source category, activity data is very uncertain and typically not recorded in statistics (e.g. domestic and industrial use of solvent products). For now, and as an alternative, emissions reported by other existing inventories will be used through the application of HERMESv3_GR. The second limitation is related to the suitability of the proposed emission estimation approaches for its application over different regions. For instance, in the case

of residential combustion emissions, HERMESv3_BU assumes that wood combustion only occurs in rural areas, which is the case for Spain. Nevertheless, this assumption may be wrong for other countries such as Norway (Lopez-Aparicio et al., 2017). Similarly, for the agriculture sector it is assumed that the temporal distribution of fertilizer application is not affected by any policy restriction (i.e. closed periods). While this is again true for Spain, there are some European countries (e.g. Germany) in which the application of fertilizer is prohibited during certain periods (Backes et al., 2016). In this sense, it is expected that the

emission approaches will be refined in future versions of HERMESv3_BU. One last aspect to consider is the availability and accessibility to all the required input data needed to run the model, which may limit the usability of the tool in certain regions. It is important to note that some of the required data can be obtained from European homogenised databases such as Eurostat (https://ec.europa.eu/eurostat/). Moreover, the continuous growth of the open data movement can also help to overcome this limitation. One example for this is the Open Transport Map (http://opentransportmap.info/), an application for accessing daily

traffic volumes for the main European road network.

HERMESv3_BU represents an effort to integrate and combine in a flexible and transparent way state-of-the-art methods for estimating high resolution bottom-up emissions from multiple anthropogenic sources. The purpose of the model is to serve as an emission tool for multiple applications, including air quality research and environmental management.



## Appendix A

**Table A1: Relationship between crop types and CORINE Land Cover (CLC) land use categories**

| Crop category | CLC code | CLC description |
|---|---|---|
| alfalfa | 12 | Non-irrigated arable land |
| almond | 16 | Fruit trees and berry plantations |
| apple | 16 | Fruit trees and berry plantations |
| apricot | 16 | Fruit trees and berry plantations |
| barley | 12 | Non-irrigated arable land |
| cherry | 16 | Fruit trees and berry plantations |
| cotton | 13 | Permanently irrigated land |
| fig | 16 | Fruit trees and berry plantations |
| grape | 15 | Vineyards |
| lemon | 16 | Fruit trees and berry plantations |
| maize | 12 | Permanently irrigated land |
| melon | 16 | Fruit trees and berry plantations |
| oats | 12 | Non-irrigated arable land |
| olive | 17 | Olive groves |
| orange | 16 | Fruit trees and berry plantations |
| pea | 12 | Non-irrigated arable land |
| peach | 16 | Fruit trees and berry plantations |
| pear | 16 | Fruit trees and berry plantations |
| potato | 13 | Permanently irrigated land |
| rice | 14 | Rice fields |
| rye | 12 | Non-irrigated arable land |
| sunflower | 12 | Non-irrigated arable land |
| tangerine | 16 | Fruit trees and berry plantations |
| tomato | 13 | Permanently irrigated land |
| triticale | 12 | Non-irrigated arable land |
| vetch | 12 | Non-irrigated arable land |
| watermelon | 16 | Fruit trees and berry plantations |
| wheat | 12 | Non-irrigated arable land |





## Appendix B

### Table B1: Classification of HERMESv3_BU input data files per pollutant source

| Sector | User-dependent files | Built-in files | External files |
|---|---|---|---|
| Point sources | • Point sources shapefile<br>• Temporal profiles CSV files<br>• Speciation profiles CSV files<br>• Meteorological files (only if plume rise is activated) (hourly 4D temperature, 4D U/V wind components, PBL height, Obukhov length, friction velocity) [1] | | |
| Road transport | • Road network shapefile<br>• Temporal profiles CSV files<br>• Fleet composition profiles CSV file | • Emission factors CSV files<br>• Speciation profiles CSV files | • ERA5 meteorological files (hourly 2-m temperature and precipitation) [2] |
| Residential & commercial combustion | • Energy consumption at NUTS3 CSV file | • Emission factors CSV files<br>• Temporal profiles CSV files<br>• Speciation profiles CSV files | • JRC global human settlement population grid<br>• JRC global human settlement city model grid<br>• ERA5 meteorological files (daily 2-m temperature) [2] |
| Shipping in ports | • Hoteling & manoeuvring shapefiles<br>• Vessel's operations CSV file | • Emission factors CSV files<br>• Vessel's technology CSV file<br>• Load factor CSV file | |
| Aviation (LTO) | • Airports, runways and air trajectories shapefiles<br>• Plane operations CSV file<br>• Temporal profiles CSV files | • Emission factors CSV files<br>• Speciation profiles CSV files | |
| Recreational boats | • Recreational boat units, load factor, working hours, nominal engine power CSV files<br>• Spatial distribution raster file | • Emission factors CSV file | |
| Livestock | • Livestock split and adjusting factors CSV file | • Emission factors CSV files<br>• Speciation profiles CSV files | • FAO gridded livestock of the world version 3<br>• ERA5 meteorological files (daily 2-m temperature and 10-m wind speed) [2] |
| Agricultural crop operations | | • Emission factors CSV file<br>• Temporal profiles CSV files<br>• Speciation profiles CSV files | • CORINE Land Cover land uses |
| Agricultural machinery | • Equipment units and nominal engine power and working hours CSV file | • Emission factors CSV files<br>• Deterioration factors CSV file<br>• Temporal profiles CSV files<br>• Speciation profiles CSV files | • CORINE Land Cover land uses |
| Agricultural fertilizers | • Fertilizer rate CSV file<br>• Crop calendar CSV file<br>• Ration of cultivated to fertilised area CSV file<br>• Share of fertilizer type per crop CSV file | • Fertilizer related emission factor parameter CSV file<br>• Temporal profiles CSV file<br>• Speciation profiles CSV file | • ISRIC soil pH and CEC data<br>• CORINE Land Cover land uses<br>• ERA5 meteorological files (daily 2-m temperature and 10-m wind speed) [2] |

[1] These meteorological parameters are not provided by ERA5 and therefore need to be derived from other models
[2] ERA5 is proposed since it is open data. Users can alternatively use the outputs from other meteorological models



## 6    Code availability

The HERMESv3_BU code package is available at the following gitlab repository: https://earth.bsc.es/gitlab/es/hermesv3_bu (doi: 10.5281/zenodo.3521897). A wiki of the model with further instructions, a detailed description of the configuration and input files needed to run the model and a test case is also included in the gitlab repository. The required libraries need to be

installed by the user in the computer infrastructure where the model is planned to be run.

## 7    Authors contribution

Marc Guevara conceived and coordinated the development of HERMESv3_BU. Marc Guevara and Manuel Porquet prepared the input databases, performed software tests and run the experiments to obtain the emission results presented. Carles Tena developed the HERMESv3_BU code and run the experiments to test the performance of the parallel implementation. Oriol

Jorba and Carlos Pérez García-Pando helped conceiving HERMESv3_BU and supervised the work. Marc Guevara prepared the manuscript with contributions from all co-authors.

## 8    Acknowledgements

The research leading to these results has received funding from Ministerio de Ciencia, Innovación y Universidades (MICINN) as part of the PAISA project CGL2016-75725-R and the BROWNING project RTI2018-099894-B-I00. Carlos Pérez García-

Pando acknowledges long-term support from the AXA Research Fund, as well as the support received through the Ramón y Cajal programme (grant RYC-2015-18690) of the Spanish Ministry of Economy and Competitiveness. The authors are thankful to the Spanish Research Centre for Energy, Environment and Technology (CIEMAT) and the Royal Automobile Club of Spain (RACC) for sharing the databases of power plants and road transport emission measurements, respectively. All the simulations were performed in the Marenostrum4 supercomputer, hosted by the Barcelona Supercomputing Center. Maps from

Fig.4.a, 5.a, 5.b, 7.a and 7.c of this manuscript were created using ArcGIS® software by Esri. ArcGIS® and ArcMap™ are the intellectual property of Esri and are used herein under license. Copyright © Esri. All rights reserved. For more information about Esri® software, please visit www.esri.com.

## 9    Competing interests

The authors declare that they have no conflict of interest.



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



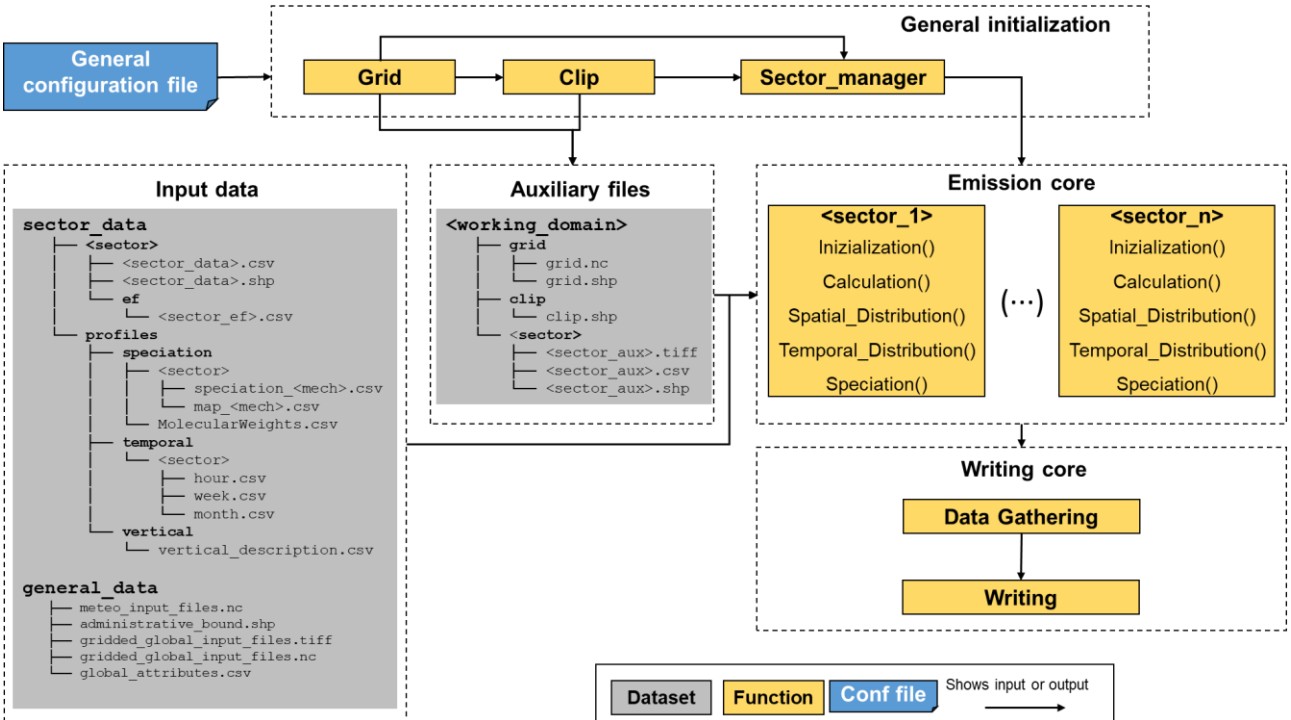

**Figure 1: Schematic representation of the general structure of HERMESv3_BU. All the parameters described by the user in the general configuration file (e.g. working domain, execution dates, input data paths) are used to execute the initialization process (i.e. creation of working grid, clip polygon and auxiliary files and distribution of the computational resources). Once the general initialization is finished, HERMESv3_BU starts the execution of the emission core of the model. For each pollutant sector, and considering the input data provided by the user, HERMESv3_BU performs a (i) sector initialization, (ii) sector calculation, (iii) spatial mapping, (iv) temporal distribution and (v) speciation. Once the emission estimation of all sectors is finished, the writing core proceeds to gather all the data and writing it according to the atmospheric chemistry output format selected by the user.**



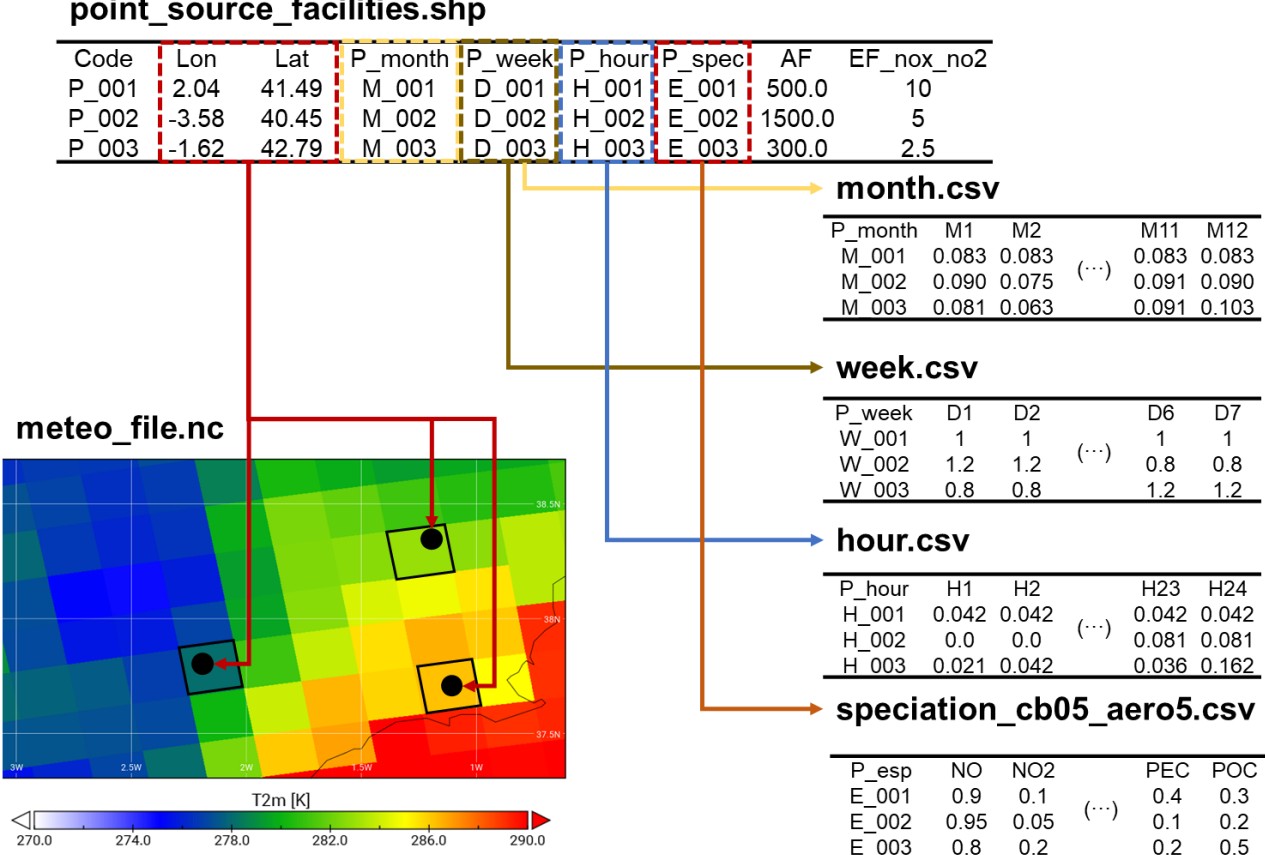

**Figure 2: Ilustration of how the different input files used by the point source sector are linked and cross-referenced. The multipoint shapefile (point_source_inventory.shp), contains temporal and speciation profile IDs for each facility (e.g. "MXXX" for the monthly profiles, "XXX" being a three-digit numeric code that starts at "001") which are cross-referenced with the corresponding temporal and speciation CSV files (month.csv, week.csv, hour.csv, speciation_cb05_aero5.csv). The shapefile also contains the geographical coordinates of each facility (Lon, Lat), which are used to identify the closest grid cell of the meteorological NetCDF file (meteo_file.nc) and derive the surface temperature data.**





**Figure 3: Examples of the spatial operations performed by HERMESv3_BU during the initialization of the residential and commercial combustion emission sector, including: a clip of the original GHSL population density raster [pop pixel$^{-1}$] using a shapefile of the administrative borders of Spain (a), a conversion of the clipped raster to a polygon feature [pop cell$^{-1}$] (zoom over the area of Madrid) (b), the spatial joins performed to append the NUTS3 administrative boundary codes (ES300, Madrid; ES424, Guadalajara and ES425, Toledo) and the GHSL settlement categories (urban, rural) to each source grid cell (c), and the spatial intersections applied to remap the fuel consumption data (natural gas in urban and rural areas and wood in rural areas) [GJ cell$^{-1}$ year$^{-1}$] from the source domain to the user-defined destination domain (4km by 4km lambert conformal conic grid).**



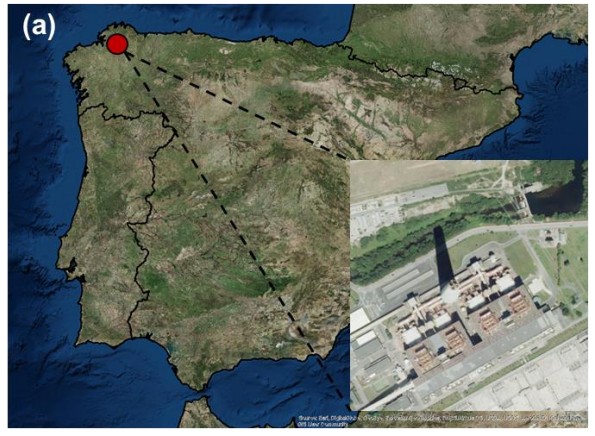

| (b) | Δh [m] | T [°C] | PBL [m] | W [m·s⁻¹] |
|---|---|---|---|---|
| January | 219.3 | 9.5 | 695.3 | 14.8 |
| July | 259.5 | 15.9 | 264.9 | 7.8 |

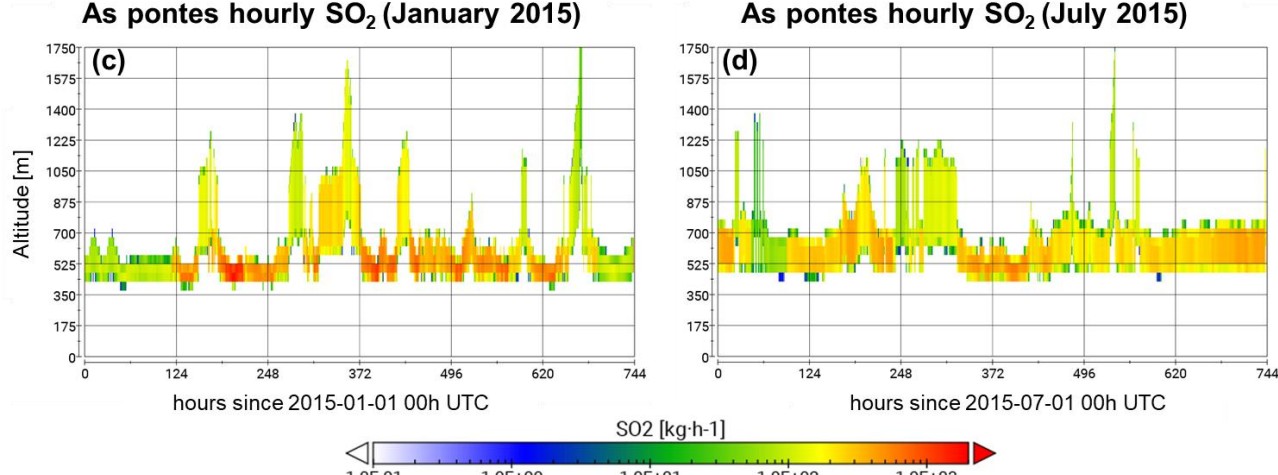

**Figure 4: (a) Location of the As Pontes Spanish coal-fired power plant (Sources of the ArcGIS Online basemap: Esri, DigitalGlobe, GeoEye, i-cubed, USDA FSA, USGS, AEX, Getmapping, Aerogrid, IGN, IGP, swisstopo, and the GIS User Community); (b) modelled average plume thickness (Δh) [m], air temperature at the stack height (T) [°C], boundary-layer height (PBL) [m] and wind speed at the stack heigh (W) [m·s⁻¹] for January and July (monthly averages); and results of the hourly and vertically distributed SO₂ emissions [kg h⁻¹] estimated by HERMESv3_BU for the months of (b) January and (c) July 2015.**





Figure 5: **Hourly PM2.5 road transport emissions estimated for an area of Barcelona city (09:00h UTC) at the road link level [kg km $^{-1}$ h$^{-1}$] (a) and grid cell level (1kmx1km) [kg h $^{-1}$] (b) (Sources of the ArcGIS Online basemap: Esri, DigitalGlobe, GeoEye, i-cubed, USDA FSA, USGS, AEX, Getmapping, Aerogrid, IGN, IGP, swisstopo, and the GIS User Community). Barcelona city total annual NO$_x$ and PM10 road transport emissions [t year$^{-1}$] estimated by HERMESv3_BU and reported by the Barcelona City Council (AB, 2015) (c).**





**Figure 6: Spanish annual NH₃ (a) fertilizer and (c) livestock emissions [t year⁻¹] calculated on a lambert conformal conic grid of 4km by 4km resolution, and corresponding time series of daily NH₃ [t day⁻¹] for the regions of (b) Aragon-Catalonia per crop type and (d) Murcia per livestock category; (e) the total NH₃ emissions (fertilizers+livestock) estimated for these two hot spots [t year⁻¹] are compared against IASI satellite-derived NH₃ emission fluxes (Van Damme et al., 2018).**





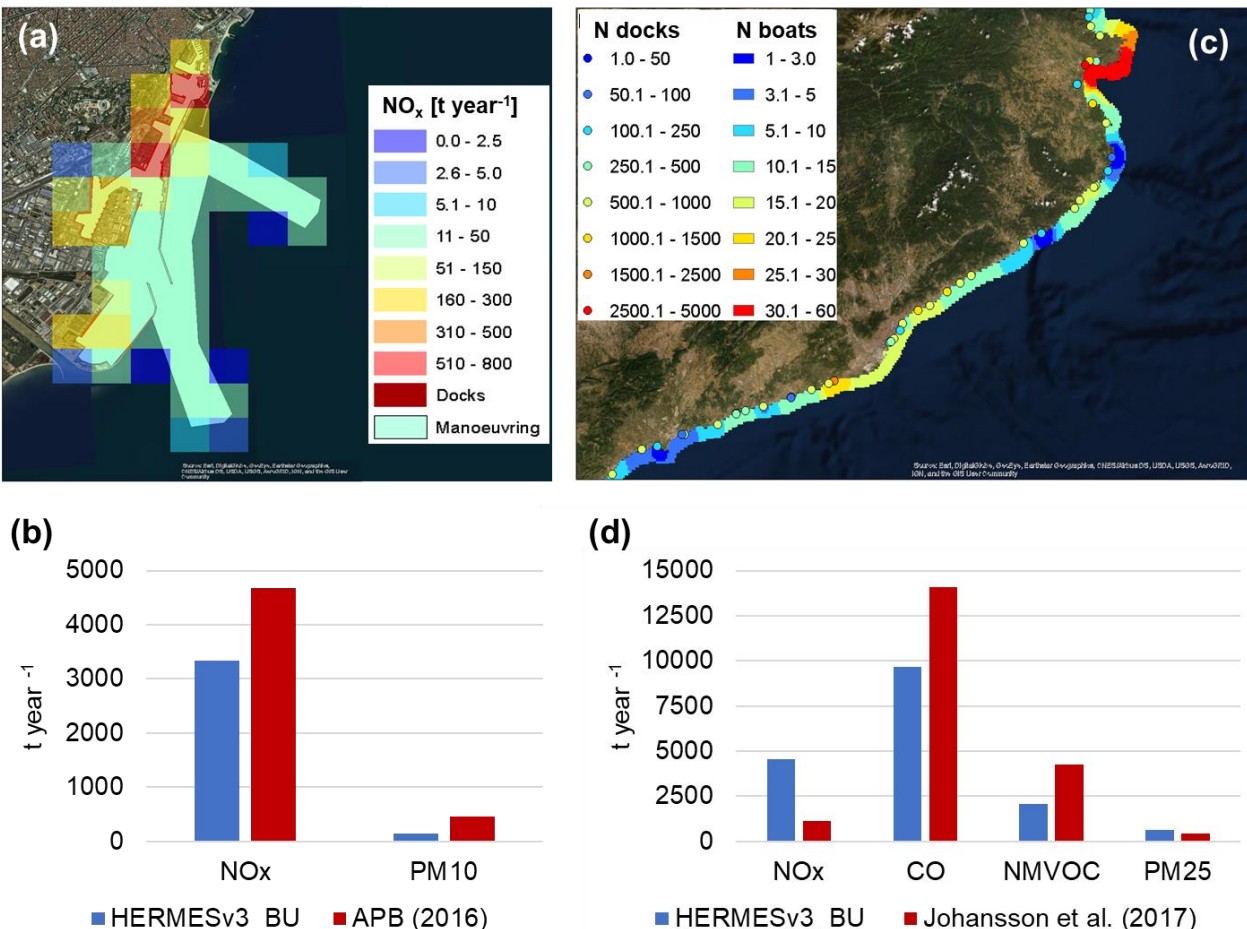

**Figure 7: (a) Annual NOₓ emissions [t year⁻¹] for the Port of Barcelona calculated on a regular 1km by 1km grid. Dark red and light blue layers represent the spatial proxies used to allocate hoteling and manoeuvring emissions, respectively; (b) Comparison between annual NOₓ and PM₁₀ Barcelona's port emissions [t year⁻¹] estimated with HERMESv3_BU and reported by the Barcelona Port Authority inventory (APB, 2016); (c) Spatial distribution of Spanish marinas (with associated number of docks) and recreational boats along the Costa Brava region (coast of Catalonia); (d) Comparison between annual NOₓ, CO, NMVOC and PM₂.₅ pleasure boat emissions [t year⁻¹] estimated for Spain (HERMESv3_BU) and the Baltic Sea region (Johansson et al., 2017). (Sources of the ArcGIS Online basemap: Esri, DigitalGlobe, GeoEye, i-cubed, USDA FSA, USGS, AEX, Getmapping, Aerogrid, IGN, IGP, swisstopo, and the GIS User Community).**



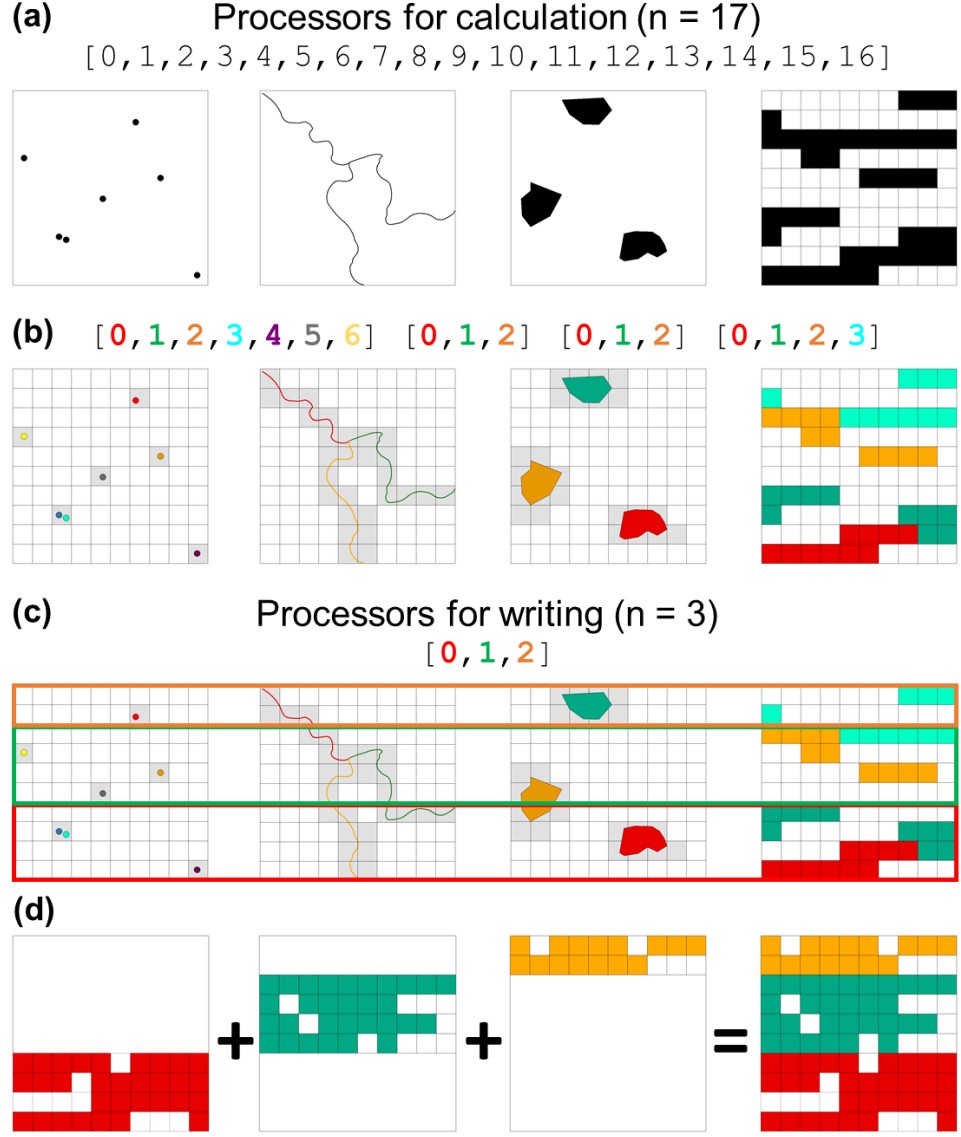

**Figure 8: Illustrative example of how HERMESv3_BU parallelises the calculation and writing processes and distributes the computational resources. (a)** A total of 17 processors are selected to estimate emissions from point sources, road transport, aviation and an area source sector (e.g. residential combustion). **(b)** The calculation processors are distributed as follows: 7 for the point sources (1 processor per facility), 3 for road transport (1 processor per road link), 3 for aviation (1 processor per polygon) and 4 for residential combustion (1 processor per subgroup of cells equally balanced, i.e. 10 cells per subgroup). Each processor estimates the emissions of the corresponding partitioned element and maps them onto the intersected grid cells (highlighted in grey). **(c)** During the writing process, HERMESv3_BU decompose the destination working domain into 3 horizontal sections (equivalent to the number of writing processors selected), maintaining each row undividable. **(d)** For each horizontal section, the model gathers the estimated gridded emissions of all the sources involved in that section and then compute the total sum per grid cell. Once this operation is finished, resulting emissions are dumped into corresponding section of the output file.





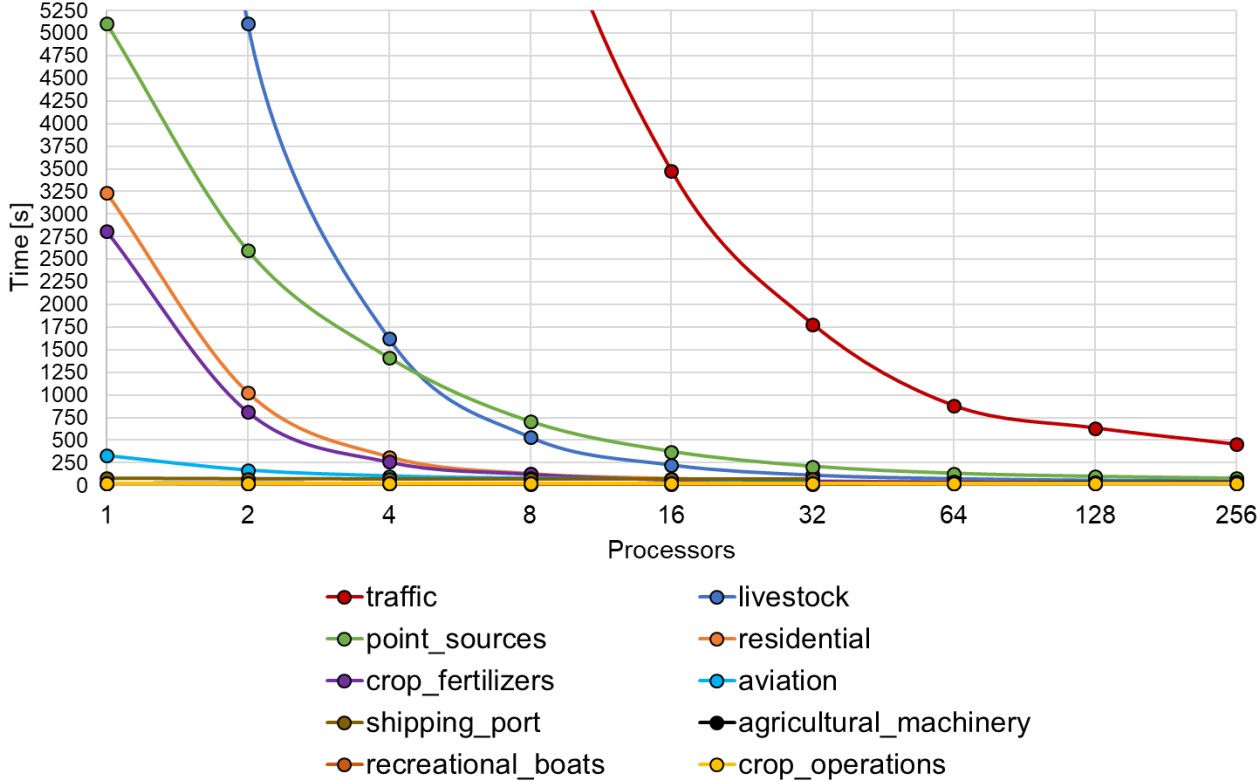

**Figure 9: Computational times [s] obtained from the scalability test performed with HERMESv3_BU. Each line represents the amount of time needed to complete the emission estimation process of each pollutant sector as a function of the number of processors used (X-axis, represented with a base 2 logarithmic scale).**

