# Peer review of "HERMESv3, a stand-alone multiscale atmospheric emission modelling framework - Part 2: bottom-up module."

_Geoscientific Model Development, 2019_

## Referee Comment (RC1) · Bok H. Baek (Referee) · 16 Dec 2019

Enjoyed reading this submitted paper. Well written and the quality of the paper is excellent. Readability was also great! It covers complex bottom-up inventory developments across significantly different types of emissions sources, such as point, transportation, agriculture and so on. Each emission sector has been well described and carefully adapted emissions calculations method from reliable sources. The authors also carefully designed the model for global modeling users on how to adapt their own emissions to use this model. These model developers understand the complexity of emissions inventory development and processing across regions/countries.

---

## Referee Comment (RC2) · Anonymous Referee #2 · 18 Jan 2020

Despite of the apparent aridity of the subject this paper is really.easy and very enjoyable to read. We learn all we need to know about the implementation of an emission bottom-up model. Every step is well documented, with relevant examples giving a clear an understandable overview of the various steps.

Somehow, one question arises: why estimated emissions for CO2 and CH4, are only related to combustion processes ?

For clarity and a better understanding is it possible to provide a diagram showing the sectors involved/available in the model, and for each sectors their major caracteristics : pollutants involved, area or point source, temporal and vertical distributions, ... Also

[Figure]

it could be useful to have a diagram showing where user-defined data are needed and therefore essential to make the model run. Add a table showing speciation.

Table B1 (Classification of HERMESv3_BU input data files per pollutant source) could be more readable if the file names appear with a color code depending on the file format (i.e. raster, shapefiles, CSV, others...)

Please also note the supplement to this comment:
https://www.geosci-model-dev-discuss.net/gmd-2019-295/gmd-2019-295-RC2-supplement.zip

---

## Author Comment (AC1) · 27 Jan 2020

Enjoyed reading this submitted paper. Well written and the quality of the paper is excellent. Readability was also great! It covers complex bottom-up inventory developments across significantly different types of emissions sources, such as point, transportation, agriculture and so on. Each emission sector has been well described and carefully adapted emissions calculations method from reliable sources. The authors also carefully designed the model for global modeling users on how to adapt their own emissions to use this model. These model developers understand the complexity of emissions inventory development and processing across regions/countries.

We are honoured and grateful for such positive review. The authors specially value referee's comments, considering that he has been the main SMOKE emission modelling system developer over 10 years. The HERMESv3 system is in its early stage, and there is still a long way to go before it may gain visibility, but the comments from Bok H. Baek inspire and encourage us to continue working and, hopefully, to make it a reference point as it is now SMOKE among the modelling community.

Anonymous Referee #2

Despite of the apparent aridity of the subject this paper is really easy and very enjoyable to read. We learn all we need to know about the implementation of an emission bottom up model. Every step is well documented, with relevant examples giving a clear an understandable overview of the various steps.

We appreciate the positive and constructive feedback of Reviewer #2, which helped improve the quality of the paper.

Somehow, one question arises: why estimated emissions for CO2 and CH4, are only related to combustion processes ?

Response to Reviewer#2 comment No. 1: In its current version, the main goal of the HERMESv3 system is to provide high-resolution emission estimations for air quality modelling. In the case of combustion sources, criteria pollutants and greenhouse gases are co-emitted species and, therefore, once you define and implement the estimation methodology approach, you can use it for all the species (i.e. you just need to change the emission factor associated with the activity). On the other hand, greenhouse gas emissions from non-combustion sources (e.g. CH4 emissions from enteric fermentation processes in the livestock sector) are governed by specific and complex processes that need to be described using specific estimation approaches, such as the ones reported in the IPCC emission inventory guidelines (e.g. https://www.ipcc-nggip.iges.or.jp/public/2006gl/pdf/4_Volume4/V4_10_Ch10_Livestock.pdf). The implementation of such processes in HERMESv3 is a task that we plan to address in future versions of the system. In order to clarify this point, we have added the following sentence in the conclusion section of the manuscript:

*"Regarding greenhouse gases, estimated emissions are currently only related to combustion processes. Emissions from non-combustion sources (e.g. CH4 emissions from enteric fermentation processes in the livestock sector) are governed by complex processes that need to be described using specific estimation approaches, such as the ones reported in the Intergovernmental Panel on Climate Change (IPCC) inventory guidelines (IPCC, 2006). The implementation of such processes in HERMESv3_BU is a task that we plan to address in future versions of the system"* (p.31, l.26 – 31 of the revised manuscript)

IPCC: 2006 IPCC Guidelines for National Greenhouse Gas Inventories, Prepared by the National Greenhouse Gas Inventories Programme, Eggleston H.S., Buendia L., Miwa K., Ngara T. and Tanabe K. (eds). Published: IGES, Japan, 2006.

For clarity and a better understanding is it possible to provide a diagram showing the sectors involved/available in the model, and for each sectors their major characteristics: pollutants involved, area or point source, temporal and vertical distributions, ...

Response to Reviewer#2 comment No. 2: Authors completely agree with the reviewer. A table (Table 1 of the revised manuscript) summarizing the main characteristics of each sector (i.e. source type, categories and processes considered, pollutants involved and temporal/vertical distribution) has been added to the revised version of the manuscript.

**Table 1: Summary of the main characteristics of each pollutant sector included in HERMESv3_BU**

| Sector | Source type | Categories and processes | Pollutants | Vertical and temporal distribution |
|---|---|---|---|---|
| Point sources | Point | Energy and manufacturing facilities and waste incinerators:
• Combustion processes
• Production processes | $NO_x$, CO, NMVOC, $SO_x$, $NH_3$, $PM_{10}$, $PM_{2.5}$, $CO_2$ and $CH_4$ | • Vertical distribution according to stack height or plume rise calculation
• Monthly, weekly and diurnal time factors |
| Road transport | Line (non-evaporative)
Area (evaporative) | COPERT 5 vehicle categories [1]:
• Exhaust (hot and cold start)
• Non-exhaust (wear, resuspension and evaporation) | $NO_x$, CO, NMVOC, $SO_x$, $NH_3$, $PM_{10}$, $PM_{2.5}$, $CO_2$ and $CH_4$ | • Ground-based emissions
• Monthly, weekly and diurnal time factors |
| Residential and commercial combustion | Area | Natural gas, liquefied petroleum gas, heating diesel oil, wood and coal:
• Combustion processes | $NO_x$, CO, NMVOC, $SO_x$, $NH_3$, $PM_{10}$, $PM_{2.5}$, $CO_2$ and $CH_4$ | • Ground-based emissions
• Day-of-year time distribution using heating degree-day approach |
| Shipping in ports | Area | EMEP/EEA (2016) ship categories [2]:
• Manoeuvring
• Hoteling | $NO_x$, CO, NMVOC, $SO_x$, $NH_3$, $PM_{10}$, $PM_{2.5}$, $CO_2$ and $CH_4$ | • Ground-based emissions
• Monthly, weekly and diurnal time factors |
| Aviation (LTO cycle) | Area | EMEP/EEA (2016) plane categories [3]:
• Land-based operations: Taxi-in, taxi-out, take-off
• Air operations: climb out and approach | $NO_x$, CO, NMVOC, $SO_x$, $NH_3$, $PM_{10}$, $PM_{2.5}$, $CO_2$ and $CH_4$ | • Vertical distribution according to LTO cycle
• Monthly, weekly and diurnal time factors |
| Recreational boats | Area | EMEP/EEA (2016) pleasure boat categories [4]:
• Combustion processes | $NO_x$, CO, NMVOC, $SO_x$, $NH_3$, $PM_{10}$, $PM_{2.5}$, $CO_2$ and $CH_4$ | • Ground-based emissions
• Monthly, weekly and diurnal time factors |
| Livestock | Area | Pigs, cattle, poultry, goats and sheep:
• Housing, yarding, storage, grazing | $NO_x$, NMVOC, $NH_3$, $PM_{10}$, $PM_{2.5}$ | • Ground-based emissions
• Day-of-year time distribution using Skjøth et al. (2011) parametrization |
| Agricultural crop operations | Area | Wheat, rye, barley and oat:
• Soil cultivation
• Crop harvesting | $PM_{10}$, $PM_{2.5}$ | • Ground-based emissions
• Monthly, weekly and diurnal time factors |
| Agricultural machinery | Area | Two-wheel tractors, agricultural tractors and harvesters:
• Combustion processes | $NO_x$, CO, NMVOC, $SO_x$, $NH_3$, $PM_{10}$, $PM_{2.5}$, $CO_2$ and $CH_4$ | • Ground-based emissions
• Monthly, weekly and diurnal time factors |
| Agricultural fertilizers | Area | Alfalfa, almond, apple, apricot, barley, cherry, cotton, fig, grape, lemon, maize, melon, oats, olive, orange, pea, peach, pear, potato, rice, rye, sunflower, tangerine, tomato, triticale, vetch, watermelon, wheat:
• Mineral fertilizers and manure application | $NH_3$ | • Ground-based emissions
• Day-of-year time distribution using Skjøth et al. (2011) parametrization |

[1] EMEP/EEA (2016) (Chapter 1.A.3.b.i-iv, Tier 3 approach)  [2] EMEP/EEA (2016) (Chapter 1.A.3.d, Tier 3 approach)  [3] EMEP/EEA (2016) (Chapter 1.A.3.a, Tier 3 approach)
[4] EMEP/EEA (2016) (Chapter 1.A.5.b, Tier 3 approach)

The table has been introduced in the text as follows:

*"Table 1 summarizes the major characteristics of each pollutant sector considered in HERMESv3_BU, including source type, categories and processes considered, pollutants involved and temporal and vertical distribution."* (p.10, l.2 – 3 of the revised manuscript)

Also it could be useful to have a diagram showing where user-defined data are needed and therefore essential to make the model run.

Response to Reviewer#2 comment No. 3: All the model input data (including user-defined, built-in and external files) are needed before the execution of the emission core of the system, as illustrated in Figure 1. Each one of the files defined in Table B1 is linked to one of the steps defined in the emission core process (initialization, calculation, spatial distribution, temporal distribution and speciation). For instance, for the case of road transport the "Fleet composition profiles" CSV file is needed for the calculation step, while the "Temporal profiles" CSV file is needed for the temporal distribution step. For clarification, we have added the following sentence in the revised version of the manuscript:

*"All the model input data (user-defined, built-in and external) are needed to correctly execute the emission core of the system, as illustrated in Figure 1."* (p.7, l.19 – 20 of the revised manuscript)

On the other hand, the data paths where the model input files are stored is an information that the user is free to define in the general configuration file (i.e. there is not a particular file data storage convention that needs to be followed). This is explained in section 2.2 of the manuscript. We have added a reference to Table B.1, to make it clearer:

*"This section contains individual subsections for each pollutant sector, in which the user defines: (i) the list of pollutants to be calculated, (ii) the data paths that point to the specific-sector input files (see Table B1) used for the emission calculation process (i.e. user can freely define its own file data storage convention) and (iii) (···)"* (p.6, l.10 – 11 of the revised manuscript)

We believe that the combination of Figure 1 + Table B.1 + the additional information introduced in the revised version of the manuscript make it no required to add an additional diagram.

Add a table showing speciation.

Response to Reviewer#2 comment No. 4: Following the reviewer's suggestion, a table (Table 2 of the revised manuscript) summarizing some examples of speciation profiles included in HERMESv3_BU for mapping emissions to CB05 and AERO5 mechanism species has been added. The example includes speciation profiles for different sector/emission categories.

Table 2 and a brief discussion of its content has been introduced in the manuscript as follows:

*"As an illustration, Table 2 shows some examples of proposed CB05 and AERO5 speciation profiles for different pollutant sectors. As mentioned before, HERMESv3_BU allows using specific profiles for each of the source categories included in the different sectors. This feature enables the user to consider key factors influencing the splitting of primary pollutants into chemical mechanism species such as: (i) fuel type (wood, natural gas) for NMVOC emissions in the residential sector (Simon et al., 2010), (ii) vehicle type (passenger car, motorcycle) and vehicle fuel (diesel, gasoline) for road transport NOx (Carslaw et al., 2016 and Rappenglueck et al., 2013), (iii) type of process (combustion, road wear) for road transport PM2.5 emissions (EMEP/EEA, 2016) (Chapter 1.A.3.b.i-iv, table 3.88 and chapter 1.A.3.b.vi, table 3-11) and (iv) animal type (pigs, cattle) for livestock NMVOC emissions (EMEP/EEA, 2016) (Chapter 3.B, table A1.2)."* (p.27, l.7 – 14 of the revised manuscript)

Table 2: Example of speciation profiles included in HERMESv3_BU for speciating primary emissions to CB05 and AERO5 mechanisms. Different pollutant sectors and categories are shown to illustrate the degree of specificity allowed by the model. The symbol "-" denotes that no primary emissions are considered for that pollutant. All CB05 and AERO5 species are defined in Table A2 of Guevara et al. (2019).

| Primary emissions | CB05 / AERO5 species | Residential & commercial combustion | | Road transport | | | | Livestock | |
|---|---|---|---|---|---|---|---|---|---|
| | | Biomass | Natural gas | Passenger Cars Petrol Euro 5 | Passenger Cars Diesel Euro 5 | Petrol Motorcycles Euro 4 | Road wear | Cattle | Pigs |
| $NO_x$ | NO | 0.9 | 0.9 | 0.97 | 0.67 | 0.96 | - | 1 | 1 |
| | NO2 | 0.1 | 0.1 | 0.013 | 0.313 | 0.023 | - | 0 | 0 |
| | HONO | 0 | 0 | 0.008 | 0.017 | 0.008 | - | 0 | 0 |
| CO | CO | 1 | 1 | 1 | 1 | 1 | - | - | - |
| $SO_x$ | SO2 | 1 | 1 | 1 | 1 | 1 | - | - | - |
| $NH_3$ | NH3 | 1 | 1 | 1 | 1 | 1 | - | 1 | 1 |
| NMVOC | PAR | 1.6E-03 | 4.5E-02 | 3.1E-02 | 2.9E-02 | 2.6E-02 | - | 3.9E-02 | 3.3E-02 |
| | OLE | 9.5E-04 | 0.0E+00 | 1.5E-03 | 1.7E-03 | 1.8E-03 | - | 2.3E-05 | 0.0E+00 |
| | TOL | 5.6E-05 | 4.9E-04 | 1.6E-03 | 3.1E-04 | 2.1E-03 | - | 2.4E-04 | 6.2E-04 |
| | XYL | 6.1E-03 | 0.0E+00 | 1.4E-03 | 8.8E-04 | 1.3E-03 | - | 0.0E+00 | 0.0E+00 |
| | FORM | 2.2E-04 | 6.1E-03 | 9.7E-04 | 4.3E-03 | 1.4E-03 | - | 0.0E+00 | 0.0E+00 |
| | ALD2 | 5.4E-05 | 0.0E+00 | 0.0E+00 | 0.0E+00 | 0.0E+00 | - | 1.6E-03 | 3.0E-03 |
| | ETH | 0.0E+00 | 0.0E+00 | 2.6E-03 | 3.9E-03 | 3.1E-03 | - | 0.0E+00 | 0.0E+00 |
| | ISOP | 0.0E+00 | 0.0E+00 | 0.0E+00 | 0.0E+00 | 0.0E+00 | - | 0.0E+00 | 0.0E+00 |
| | MEOH | 0.0E+00 | 0.0E+00 | 0.0E+00 | 0.0E+00 | 0.0E+00 | - | 0.0E+00 | 0.0E+00 |
| | ETOH | 0.0E+00 | 0.0E+00 | 0.0E+00 | 0.0E+00 | 0.0E+00 | - | 2.2E-05 | 0.0E+00 |
| | ETHA | 0.0E+00 | 0.0E+00 | 1.1E-03 | 1.1E-04 | 5.5E-04 | - | 0.0E+00 | 0.0E+00 |
| | IOLE | 0.0E+00 | 0.0E+00 | 3.0E-04 | 9.3E-05 | 2.6E-04 | - | 0.0E+00 | 0.0E+00 |
| | ALDX | 5.6E-04 | 0.0E+00 | 1.2E-04 | 1.7E-03 | 2.5E-04 | - | 1.0E-04 | 1.0E-03 |
| | TERP | 0.0E+00 | 0.0E+00 | 0.0E+00 | 0.0E+00 | 0.0E+00 | - | 7.3E-06 | 0.0E+00 |
| | BENZENE | 0.0E+00 | 1.2E-03 | 7.2E-04 | 2.5E-04 | 8.7E-04 | - | 3.8E-05 | 2.6E-05 |
| $PM_{2.5}$ | POC | 0.43 | 0.49 | 0.45 | 0.525 | 0.625 | 0.135 | 0 | 0 |
| | PEC | 0.07 | 0.067 | 0.15 | 0.15 | 0.25 | 0.0106 | 0 | 0 |
| | PNO3 | 0 | 0 | 0 | 0 | 0 | 0 | 0 | 0 |
| | PSO4 | 0 | 0 | 0 | 0 | 0 | 0 | 0 | 0 |
| | PMFINE | 0.5 | 0.443 | 0.4 | 0.325 | 0.125 | 0.8544 | 1 | 1 |

Table B1 (Classification of HERMESv3_BU input data files per pollutant source) could be more readable if the file names appear with a color code depending on the file format (i.e. raster, shapefiles, CSV, others…)

Response to Reviewer#2 comment No. 5: Authors completely agree with the reviewer. Following the suggestion, a color code has been added to the table to describe the file format of each dataset.

| Sector | User-dependent files | Built-in files | External files |
|---|---|---|---|
| Point sources | • Point sources [1]
• Temporal profiles
• Speciation profiles
• Meteorological files (only if plume rise is activated) (hourly 4D temperature, 4D U/V wind components, PBL height, Obukhov length, friction velocity) [2] | | |
| Road transport | • Road network
• Temporal profiles
• Fleet composition profiles | • Emission factors
• Speciation profiles | • ERA5 meteorological files (hourly 2-m temperature and precipitation) [3] |
| Residential & commercial combustion | • Energy consumption at NUTS3 | • Emission factors
• Temporal profiles
• Speciation profiles | • JRC global human settlement population grid
• JRC global human settlement city model grid
• ERA5 meteorological files (daily 2-m temperature) [3] |
| Shipping in ports | • Hoteling & manoeuvring
• Vessel's operations | • Emission factors
• Vessel's technology
• Load factor | |
| Aviation (LTO) | • Airports, runways and air trajectories
• Plane operations
• Temporal profiles | • Emission factors
• Speciation profiles | |
| Recreational boats | • Recreational boat units, load factor, working hours, nominal engine power
• Spatial distribution | • Emission factors | |
| Livestock | • Livestock split and adjusting factors | • Emission factors
• Speciation profiles | • FAO gridded livestock of the world version 3
• ERA5 meteorological files (daily 2-m temperature and 10-m wind speed) [3] |
| Agricultural crop operations | | • Emission factors
• Temporal profiles
• Speciation profiles | • CORINE Land Cover land uses |
| Agricultural machinery | • Equipment units and nominal engine power and working hours | • Emission factors
• Deterioration factors
• Temporal profiles
• Speciation profiles | • CORINE Land Cover land uses |
| Agricultural fertilizers | • Fertilizer rate
• Crop calendar
• Ration of cultivated to fertilised area
• Share of fertilizer type per crop | • Fertilizer related emission factor parameter
• Temporal profiles
• Speciation profiles | • ISRIC soil pH and CEC data
• CORINE Land Cover land uses
• ERA5 meteorological files (daily 2-m temperature and 10-m wind speed) [3] |

[1] Colours are used to specify the file format of each dataset: shapefile CSV NetCDF Raster
[2] These meteorological parameters are not provided by ERA5 and therefore need to be derived from other models
[3] ERA5 is proposed since it is open data. Users can alternatively use the outputs from other meteorological models

Please also note the supplement to this comment: https://www.geosci-model-dev-discuss.net/gmd-2019-295/gmd-2019-295-RC2- supplement.zip

Response to Reviewer#2 comment No. 6: We have edited the text in the manuscript according to all the reported comments.

---

## Author Response (AR2)

Thank you for accepting the manuscript. The minor comment of the editor is indicated in the following text. Modifications are highlighted with track changes in the revised version of the manuscript.

**Editor**

5 **Comments to the Author:**

Congratulations!

One minor comment to the authors: In the new table 2, several NMVOC profiles sum to much less than one. Is there nonreactive mass or are profiles not designed to sum to one? Is speciation by mass or mole in Table 2?

NMVOC speciation factors are mol-based (mol of chemical species · g of source pollutant$^{-1}$) and therefore they are not designed to sum to one. On the contrary, speciation factors for the other primary pollutants (i.e. NOx, CO, PM2.5, NH3, NH3) are mass-based (i.e. g of chemical species · g of source pollutant$^{-1}$). This information is included in section 3.6 but as pointed out by the editor it is missing in the caption of the new Table 2. We have added the following sentence in Table2 caption:

*"Speciation factors are mole-based (mol of chemical species · g of source pollutant$^{-1}$) for NMVOC and mass-based (g of chemical species · g of source pollutant$^{-1}$) for $NO_x$, CO, $SO_x$, $NH_3$ and $PM_{2.5}$."*

Besides answering the editor's comment, authors have also updated one of the reference:

Original reference:

[revised manuscript text omitted]